



# Chemical and visual characterisation of EGRIP glacial ice and cloudy bands within

Nicolas Stoll[1,2], Julien Westhoff[3], Pascal Bohleber[4], Anders Svensson[3], Dorthe Dahl-Jensen[3,5], Carlo Barbante[4,6], and Ilka Weikusat[1,7]

[1]Department of Geosciences, Alfred Wegener Institute Helmholtz Centre for Polar and Marine Research, Bremerhaven, Germany
[2]Department of Geosciences, University of Bremen, Bremen, Germany
[3]Physics of Ice, Climate, and Earth, Niels Bohr Institute, Copenhagen, Denmark
[4]Department of Environmental Sciences, Informatics and Statistics, Ca'Foscari University of Venice, Venice, Italy
[5]Centre for Earth Observation Science, University of Manitoba, Winnipeg, Canada
[6]Institute of Polar Sciences, CNR, Venice, Italy
[7]Geoscience Department, Eberhard Karls University, Tübingen, Germany

**Correspondence:** Nicolas Stoll (nicolas.stoll@awi.de)

**Abstract.** Impurities in polar ice play a critical role in ice flow, deformation, and the integrity of the ice core record. Especially cloudy bands, visible layers with high impurity concentrations are prominent features in ice from the last glacial. Their physical and chemical properties are poorly understood, highlighting the need to analyse them in more detail. We bridge the gap between decimetre and micrometre scales by combining the visual stratigraphy line scanner, fabric analyser, microstructure mapping,

Raman spectroscopy, and laser ablation inductively coupled plasma mass spectrometry 2D impurity imaging. We classified almost 1300 cloudy bands from glacial ice from the East Greenland Ice-core Project (EGRIP) ice core into seven different types. We determine the localisation and mineralogy of more than 1000 micro-inclusions at 13 depths. The majority of the found minerals are related to terrestrial dust, such as quartz, feldspar, mica, and hematite. We further found carbonaceous particles, dolomite, and gypsum in high abundance. Rare minerals are e.g., rutile, anatase, epidote, titanite, and grossular. 2D

impurity imaging with 20 $\mu$m resolution revealed that Na, Mg and Sr are mainly at grain boundaries. Dust-related analytes, such as Al, Fe, and Ti, are also located in the grain interior forming clusters of insoluble impurities. Cloudy bands are thus clearly distinguishable in the chemical data. We present novel vast micron-resolution insights into cloudy bands and describe the differences within and outside these bands. Combining the visual and chemical data results in new insights into the formation of different cloudy band types and could be the starting point for future in-depth studies on impurity signal integrity and internal

deformation.

## 1  Introduction

Deep ice cores from the polar regions revealed vast amounts of information regarding the climate of the past and processes taking place inside the ice. Pioneering deep ice cores were drilled almost 70 years ago at Camp Century, Greenland or at Byrd Station, Antarctica. Over the last decades, a variety of locations in Greenland (e.g., Greenland Ice Sheet Project Two (GISP2),



Eemian interglacial in the North Greenland Eemian Ice Drilling (NEEM), North Greenland Ice Core Project (NGRIP)) and Antarctica (e.g., West Antarctic Ice Sheet (WAIS), European Project for Ice Coring in Antarctic Dome Concordia (EPICA Dome C), Dome Fuji) were chosen for drilling operations. Depending on the drilling location and the investigated depth regimes properties, such as insoluble particle content, crystal preferred orientation, grain size, and water isotope signal, vary. However, some specific features were observed in all ice cores reaching the last glacial: the so-called "cloudy bands".

Cloudy bands are characterised by small ice crystals with a very high concentration of micro-inclusions and other impurities (e.g., Ram and Koenig, 1997; Barnes et al., 2002; Svensson et al., 2005; Faria et al., 2010; Eichler et al., 2017). Gow and Williamson (1971, 1976) were among the first to describe cloudy bands in the Byrd ice core. They observed two distinct bands: dirt bands and the much more abundant cloudy bands. Dirt bands contained large particles, detectable by the eye, and were classified as volcanic ash bands. Cloudy bands, however, were not composed of visible debris but of a greyish-white

appearance, hence the name (Gow and Williamson, 1971, 1976). Gow and Williamson (1976) report that cloudy bands are between 1 and 60 mm thick, grains inside are often smaller than 5 $\mu$m, and their preferred crystal orientation is clustered about the vertical indicating strong horizontal shearing. Cloudy bands were thus associated with dust and deformation and interpreted as shear bands (Gow and Williamson, 1971, 1976).

There is no typical cloudy band (Winstrup et al., 2012), but they have been discussed for a variety of reasons, ranging from

climatic to deformation aspects (e.g., Svensson et al., 2005; Andersen et al., 2006; Faria et al., 2010; Winstrup et al., 2012; Westhoff et al., 2021). Svensson et al. (2005) show that, in most cases, the brightness variations of cloudy bands match the seasonal cycles of other tracers by comparing the brightness intensity values derived from the visual stratigraphy with different records from continuous flow analysis (CFA). Regularly appearing cloudy bands from visual stratigraphy were further used for dating of the last glacial (Andersen et al., 2006). Cloudy bands showing visible evidence of stratigraphic disturbances and

even folding were declared as the most significant optical stratigraphy feature helping to examine the integrity of deep ice cores, thus impacting scales larger than their size (Faria et al., 2010). The fine grains within cloudy bands could enable a more efficient diffusion due to the higher availability of impurity diffusion paths, such as veins along triple junctions and planes and interfaces along grain boundaries (Faria et al., 2010). Faria et al. (2010) concluded that cloudy bands are important for the disturbance of stratigraphic records on the micro-scale, indicating anomalies in the ice rheology. Impurity-enhanced ice flow in

cloudy bands could impact ice core dating by enabling heterogeneous layer thinning (e.g., Paterson, 1991; Faria et al., 2006). Compression tests indicated that cloudy bands increase the flow enhancement factor in the flow law description by increasing the ratio of the observed strain rate to the strain rate produced on isotropic ice under the same stresses, and thus affect the bulk deformation rate of ice, i.e. soften it (Miyamoto et al., 1999). In these particular layers, microshear, i.e., the enhancement of dislocation creep by an accommodating mechanism involving grain boundary sliding (Kuiper et al., 2020) and microshear

boundary formation (Bons and Jessell, 1999; Faria et al., 2006), could be a relevant microstrain mechanism. Developing a better understanding of the interplay between impurities and the microstructure in cloudy bands is thus necessary for a holistic understanding of deformation in polar ice (Stoll et al., 2021b).

Studies suggesting particulate matter as the main reason for the visibility of cloudy bands (Svensson et al., 2005; Faria et al., 2010) opposed speculations about their appearance due to micro-bubbles forming around impurities (Dahl-Jensen et al.,





1997; Shimohara et al., 2003). Visual stratigraphy, dust, and Ca-concentration correlate well in the NGRIP ice core (Svensson et al., 2005), similar to results of 90° laser-light scattering and dust concentration in the GISP2 ice core (Ram and Koenig, 1997). Svensson et al. (2005) explain cloudy bands by the increased transport of mainly insoluble dust to the ice sheets. Each cloudy band thus represents a deposition event, i.e. precipitation or wind-driven sastrugi formation. Thin and bright bands could originate from low precipitation-dry deposition events or strong and early scavenging during snowfalls (Svensson et al.,

60 2005).

High-resolution microstructural data are needed to truly investigate the origin and the chemistry and localisation of impurities in cloudy bands. Della Lunga et al. (2017) analysed a cloudy band with laser ablation inductively coupled plasma mass spectrometry (LA-ICP-MS), but the implementable resolution limited microstructural insights. Recent methodological progress now enables state-of-the-art LA-ICP-MS 2D chemical imaging of the total impurity content at the scale of a few

ten microns (Bohleber et al., 2020, 2021), in particular when focusing also on dust-related elemental species (Bohleber et al., 2022). Furthermore, cryo-Raman spectroscopy coupled with microstructure-mapping was established as a powerful tool to localise and identify solid micro-inclusions in the microstructure of ice (Eichler et al., 2017, 2019; Stoll et al., 2021a, 2022). Hence, the mineralogy and localisation of micro-inclusions in the upper 1340 m, i.e. the Holocene, Younger Dryas, and the Bølling–Allerød, of the East Greenland Ice-core Project (EGRIP) ice core were recently analysed (Stoll et al., 2021a, 2022).

Micro-inclusions are mainly in the grain interior but show a strong heterogeneity in distribution. The principal minerals are gypsum, quartz, feldspar, and mica and mineral diversity decreases slightly with depth. However, various sulfate-minerals, such as Mg-/Na-/K- sulfates or bloedite, were found in the upper 900 m. In addition, visual stratigraphy was measured continuously on the EGRIP ice core (Westhoff et al., 2021, 2022).

This study aims to investigate EGRIP glacial ice with different methods covering several spatial scales, ranging from decime-

tres to micrometres, enabling a holistic analysis. First, we analyse the grain size evolution with depth and classify different cloudy bands and their abundance throughout the glacial using visual stratigraphy data. We explore different cloudy band types and discuss possible origins. To increase our understanding of impurity-related processes in ice, we locate and identify the mineralogy of more than 1000 micro-inclusions throughout EGRIP glacial ice using Raman spectroscopy again focusing on cloudy bands. To explore the future possibilities of inter-method studies we conduct LA-ICP-MS 2D chemical imaging on a

subset of these previously analysed samples to obtain spatial information on the major, soluble and insoluble, elements, such as Na, Mg, Sr, Al, Ti and Fe. Together, this results in a detailed study of the chemical and visual properties of EGRIP glacial ice with an emphasis on cloudy bands.

## 2 Methods

### 2.1 The East Greenland Ice Core Project

EGRIP is a deep ice core drilling project located on the Northeast Greenland Ice Stream (NEGIS), the largest ice stream in Greenland (Fahnestock et al., 1993; Vallelonga et al., 2014). The drill site is located at 75°37.820 N and 35°59.556 W, 2704 m



a.s.l, 440 km to the South-East of the NEEM site. The ice flow velocity at the drill site is ~55 m/yr (Hvidberg et al., 2020). To date, 2418 m of ice has been drilled and partly processed, with ~250 m remaining to bedrock.

## 2.2 Grain size measurements

Grain size was measured at EGRIP on discrete samples every 5-15 m of depth. 55 cm samples were cut into six samples (parallel to the axis of the core) with dimensions of ~90 x 70 x 0.3 mm. Samples were polished and measured with an automated G50 automatic fabric analyser (Russell-Head type (Wilson et al., 2003)). Details about the procedure and data processing are found in Stoll et al. (2021a).

**Table 1.** Different cloudy band types in EGRIP last glacial ice. Image width is 7 cm.

| Category | Visual example | Description | Amount (%) |
|---|---|---|---|
| Single thin cloudy band (single) | | Single thin cloudy band with dark layers above and below | 29.4 |
| Bright layer at top (up) | | Bright layer, followed by intermediate gray layer(s) | 14.7 |
| Brighter layer in the centre (centre) | | Bright layer with gray layers | 7.8 |
| Brighter layer at the bottom (bottom) | | Intermediate grey layers followed by bright layer, then dark layer | 16.9 |
| Brighter layer at top and bottom (confined) | | Central part of cloudy band is darker than the confining upper and lower boundary | 8.9 |
| Homogeneous (homogeneous) | | Homogeneous gray colour with little variation | 3.0 |
| Heterogeneous (heterogeneous) | | Combinations not fitting any category | 9.7 |
| Unknown | - | Not distinguishable | 9.6 |





## 2.3 Visual stratigraphy

Visual stratigraphy measurements, i.e. line scans, were conducted in the EGRIP trench. Svensson et al. (2005) showed the importance of performing visual stratigraphy measurements as soon as possible after ice core retrieval. Measurements were thus conducted shortly after the retrieval of the core, with some buffer time in between to account for differences in atmospheric pressure. Only ice from the brittle zone was measured a year after retrieval to lower the risk of ice breaking during processing.

Processed line scan data are available from 13.75 m to 2120 m of depth (Weikusat et al., 2020). The instrument used was a Schäfter+Kirchhoff GmbH Line Scanner, developed in cooperation with the Alfred Wegener Institute Helmholtz Centre for Polar and Marine Research (AWI) and the University of Copenhagen (details in e.g., Svensson et al. (2005); Faria et al. (2018); Westhoff et al. (2022)). Slabs of ice were polished from both sides and illuminated from below ("dark field" imaging). The light is reflected by solid impurities, fractures, and bubbles and travels freely through clean ice. A camera scans the surface during the illumination process detecting areas of reflected ice and making them visible. In the last glacial period, the main reflective objects are cloudy bands and fractures. A detailed description of the measurement process is given by Faria et al. (2018); Westhoff et al. (2021).

### 2.3.1 Cloudy band types

To compare cloudy bands throughout the ice core, a consistent camera setting is of uttermost importance. Below a depth of 1375 m (bag 2500, 14.6 ka b2k (Gerber et al., 2021)) constant settings were used. We do not investigate cloudy bands from the Holocene (Westhoff et al., 2022) and cloudy bands showing deformation features to reduce one factor of complication.

Andersen et al. (2006) and Winstrup et al. (2012) showed that cloudy bands appear in an annual cyclicity and that a single cloudy band, i.e. a stack of bright layers, is situated between two dark layers. We thus define a cloudy band by an upper and lower dark layer boundary, i.e. all bright layers between two dark layers are defined as one cloudy band. We identified seven different types of cloudy bands, designed to be specific identifiers, i.e. mutually exclusive classes, depending on where the brightest layer of one cloudy band is situated (Table 1). We find thin single bright layers (*single*), bright layers with an even brighter layer at the top (*up*), in the middle (*centre*), or in the bottom (*bottom*). We identify two bright layers confining a less bright layer at the top and bottom (*confined*), a thick bright layer with very little brightness variations (*homogeneous*), or a mix of the above, mostly with thin alternating layers (*heterogeneous*). Some layers cannot be clearly grouped (*unknown*), either because the images are too dark, the layers too thin, or features are folded. Our cloudy band types are also visible in the NEEM and NGRIP ice cores and are thus representative of Greenland.

### 2.3.2 Grayscale

The grayscale analysis investigates brightness variations caused by the scattering of light by features in the ice core. Variations occur on the cm-scale (cloudy bands) and the m-scale (Stadials-Interstadials). For the depth of investigation, we analyze the brightness derived from the pixel values of the line scan grayscale images. Values are between 0 and 255 providing 256



possibilities. One centimetre in length is equivalent to 186 pixels in the image files generating a high-resolution depth series of brightness variations.

## 2.4 Raman spectroscopy

The remaining pieces of the fabric analyser samples were cut into cubes of ca. 2 x 2 cm (Table 2) and polished from two sides with a microtome to enable successful microstructure mapping (Stoll et al., 2021a) and Raman spectroscopy analyses (Stoll
et al., 2022).

Raman spectroscopy measurements were conducted in a cold lab (-17°C) at AWI in Bremerhaven, Germany. The spectrometer, excitation laser, and control unit are located at room temperature close to the cold lab, which contains the microscope unit. A 100 $\mu$m fibre was used for good signal intensity and confocality. We used a WITec alpha300 M+ combined with a Nd:YAG laser (lambda=532 nm) and a UHTS 300 spectrometer with a 600 grooves mm$^{-1}$ grating. The pixel resolution is
<3 cm$^{-1}$ and the spectral range is >3700 cm$^{-1}$. A Hg/Ar spectral calibration lamp was used to calibrate the system. Spectra were background corrected and identified using the RRUFF database (Lafuente et al., 2015) and reference spectra (e.g., Eichler et al., 2019; Stoll et al., 2022).

## 2.5 Laser ablation inductively coupled plasma mass spectrometry 2D impurity imaging

Following Raman spectroscopy analysis at AWI the samples were transported via a commercial freezer transport to Ca'Foscari
University of Venice. Micron-resolution LA-ICP-MS 2D imaging was performed with a set-up comprised of an Analyte Excite ArF excimer 193 nm laser (Teledyne CETAC Photon Machines) with a HelEx II two-volume ablation chamber. The ablated material is transported to an iCAP-RQ quadrupole ICP-MS (Thermo Scientific) via a rapid aerosol transfer line. Samples surfaces are cleaned and polished with ceramic $ZrO_2$ blades (American Cutting Edge, USA), and the sample is then placed on a cryogenic sample holder. Before each measurement, 1-2 pre-ablation runs are conducted with an 80 x 80 $\mu$m square spot.
Before and after each measurement one scan line on a standard (NIST glass SRM 612) is acquired. We used laser spot sizes of 35 and 20 $\mu$m on the following samples analysed before with Raman spectroscopy: S2, S4, S7, S8, S10, S11, and S12 (Table 2). We followed a newly refined multi-element imaging method including dust-related elements as described in Bohleber et al. (2022). In order to consider species with mostly soluble as well as mostly insoluble behaviour, we focused on the following analytes: $^{23}$Na, $^{24}$Mg, $^{27}$Al, $^{48}$Ti, $^{56}$Fe and $^{88}$Sr.

## 3  Results

### 3.1  Grain size

Below 1340 m, mean grain size values of 9 cm samples (equals between 471 and 4550 grains per sample) generally range from 1 to 2 mm$^2$ (Fig. 1). Grains are within this size regime until 1800 m, except at 1600 m, where mean grain sizes are between 2 and 2.7 mm$^2$. Until 2121 m, values increase and spread with depth and are between 0.8 and 5.2 mm$^2$.



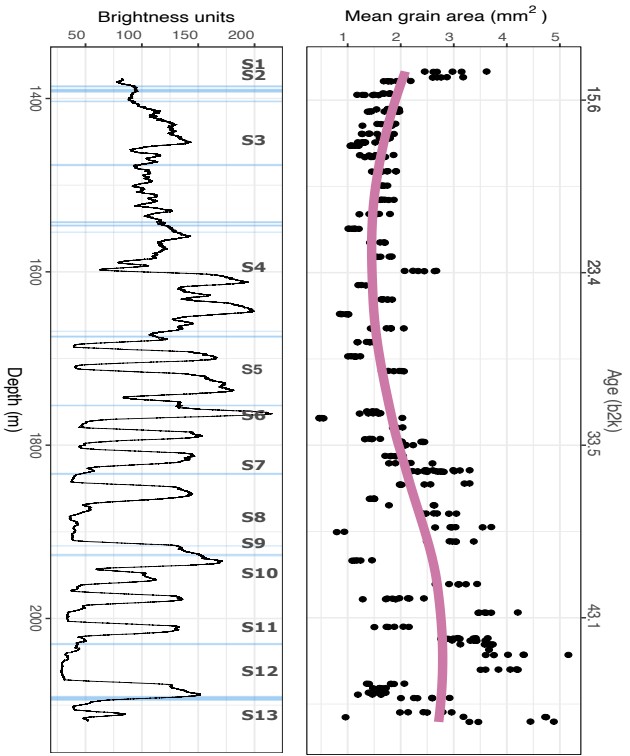

**Figure 1.** Grayscale with depth smoothed with a 5 m running mean. Horizontal light blue lines indicate samples analysed for cloudy band types; samples analysed with Raman spectroscopy are indicated in bold (S1-S13) (exact depths in Table 2). Mean grain areas of 9 cm samples from the G50 fabric analyser. The violet line is a locally weighted regression with a smoothing parameter of 0.3. Age from Gerber et al. (2021).

## 3.2 Visual stratigraphy

### 3.2.1 Grayscale

The grayscale analysis (Fig. 1), i.e. the brightness variations of line scan images, shows fluctuations following the glacial Stadials and Interstadials (Rasmussen et al., 2006). For the NGRIP ice core, the brightness curve correlates well with the measured solid particle concentration, the brightness variations can thus be used a proxy for solid particle concentration (Svensson et al., 2005). However, the relationship between particle size and grayscale is unexplored.

### 3.2.2 Types of cloudy bands

We classified 1267 cloudy bands into 7 types (Fig. 2). The type *single* is the most abundant one making up 29% of all cases (Table 1). The types *up* and *bottom*, i.e. a bright layer at the top or bottom of a homogeneous layer, add up to 15% and 17%,





respectively. The types *confined* (9%), *heterogeneous* (10%), and *centre* (8%) make up another third of all cloudy band types.
Rarest is the thick *homogeneous* (3%) type. Almost 10% could not be distinguished clearly (*unknown*).

Throughout the analysed depth, the relative distribution of cloudy bands varies per 55 cm section, especially for *single*, *homogeneous* and *heterogeneous* (Fig. 2a). Type *single* cloudy bands occur more often with depth and range from 5% to above 50% of identified types per 55 cm ice core. The types *up*, *centre*, and *bottom* are fairly constant. The type *unknown* occurs more frequently in deeper ice (Fig. 2a) as layers thin and categorisation becomes more difficult. Furthermore, the darkness of
the image plays a role. For consistency, throughout the glacial, the brightness is kept constant for all images, thus making our analysis more favourable for Stadials, i.e. cold periods with a higher dust concentration, within the last glacial period.

We identified 989 and 278 cloudy bands in Greenland Stadials and Interstadials, respectively. *Single* dominates both period types (if identifiable) with a similar relative abundance (Fig. 2b). Especially *bottom*, *heterogeneous* and *confined* cloudy bands are more common in Stadial ice. However, 27.3% of cloudy bands in Interstadials could not be identified clearly (type *unknown*)
compared to 4.6% in Stadials.

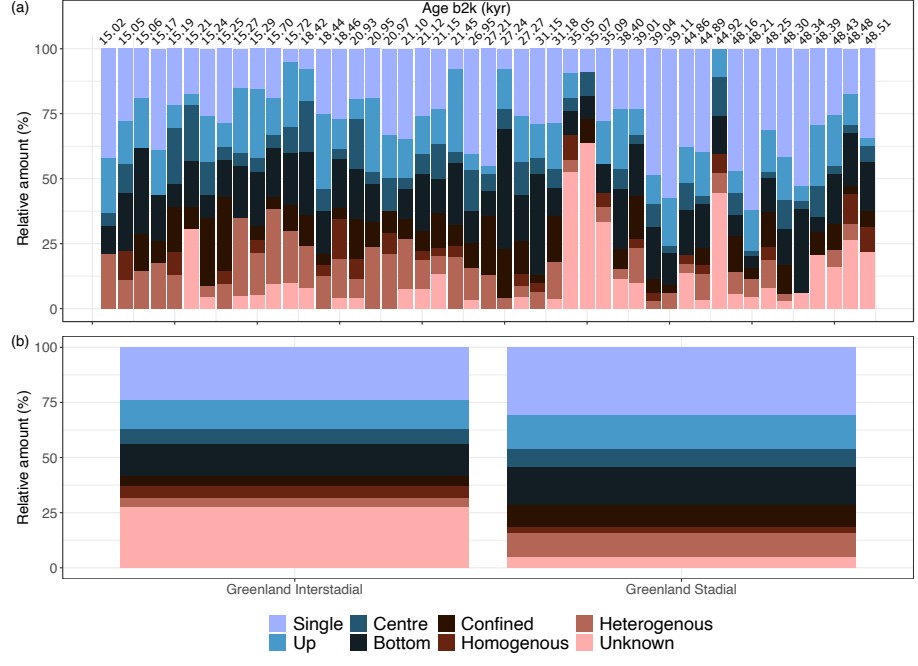

**Figure 2.** Different cloudy band types in EGRIP glacial ice. a) Cloudy band types per 55 cm ice cores (depths are shown in Fig. 1) containing identifiable cloudy bands between 15.02 and 48.51 ka before 2000 CE (Gerber et al., 2021). The different types seen in the visual stratigraphy data are described in Table 1. b) Relative amounts of all classified cloudy band types in our samples found in either Greenland Interstadials or Stadials (after Rasmussen et al. (2014)).





**Table 2.** Samples analysed with Raman spectroscopy and LA-ICP-MS (x). The depth refers to the middle of the area analysed with Raman spectroscopy.

| Sample | Depth (m) | Age b2k (ka) | Size (mm x mm) | Number of identified spectra | LA-ICP-MS |
|--------|-----------|--------------|----------------|------------------------------|-----------|
| S1 | 1360.82 | 14.4 | 17.13 x 17.10 | 117 | |
| S2 | 1367.05 | 14.5 | 16.00 x 17.28 | 130 | x |
| S3 | 1448.78 | 17.4 | 14.36 x 15.30 | 86 | |
| S4 | 1597.89 | 23.3 | 13.92 x 14.64 | 105 | x |
| S5 | 1713.84 | 28.9 | 15.73 x 19.67 | 113 | |
| S6 | 1768.34 | 32.0 | 20.82 x 25.44 | 69 | |
| S7 | 1823.48 | 34.7 | 13.97 x 20.02 | 45 | x |
| S8 | 1883.06 | 37.3 | 15.94 x 17.91 | 36 | x |
| S9 | 1917.07 | 38.5 | - | 52 | |
| S10 | 1949.98 | 39.9 | 18.15 x 15.11 | 112 | x |
| S11 | 2015.98 | 44.0 | 18.88 x 19.71 | 84 | x |
| S12 | 2065.20 | 46.6 | 15.24 x 20.92 | 50 | x |
| S13 | 2115.07 | 49.8 | 20.3 x 27.14 | 52 | |

b2k: before 2000 CE (Gerber et al., 2021). In S9, a specific cloudy band was analysed, but the sample was not cut into specific dimensions for Raman analysis.

## 3.3 Raman spectroscopy

### 3.3.1 Identified minerals

In our 13 samples, we measured 1089 spectra and identified 1051 of them (Fig. 3) resulting in 23 different Raman spectra. 188 micro-inclusions showed luminescence. The chemical formulas of all found minerals are displayed in Table A1.

The most common mineral is quartz (n=268), followed by carbonaceous particles (n=170), and the sulfate mineral gypsum (n=134). Other sulfate minerals are hexahydrite (n=7), Na and/or Mg-sulfate (n=4), bloedite (n=1), and undefined sulfates (n=15). Further minerals are feldspar (n=119), mica (n=92), hematite (n=86), calcite (n=62), K-nitrates (n=30), dolomite (n=25), magnetite (n=11), and rutile (n=10). We also identified air (n=5), titanite (n=3), anatase (n=2), and epidote (n=2). Minerals, which have not been identified before in ice cores, are whitlockite (n=2), grossular (n=1), datolite (n=1), and pumpellyite

(n=1); reference and observed spectra are displayed in Fig. A2.

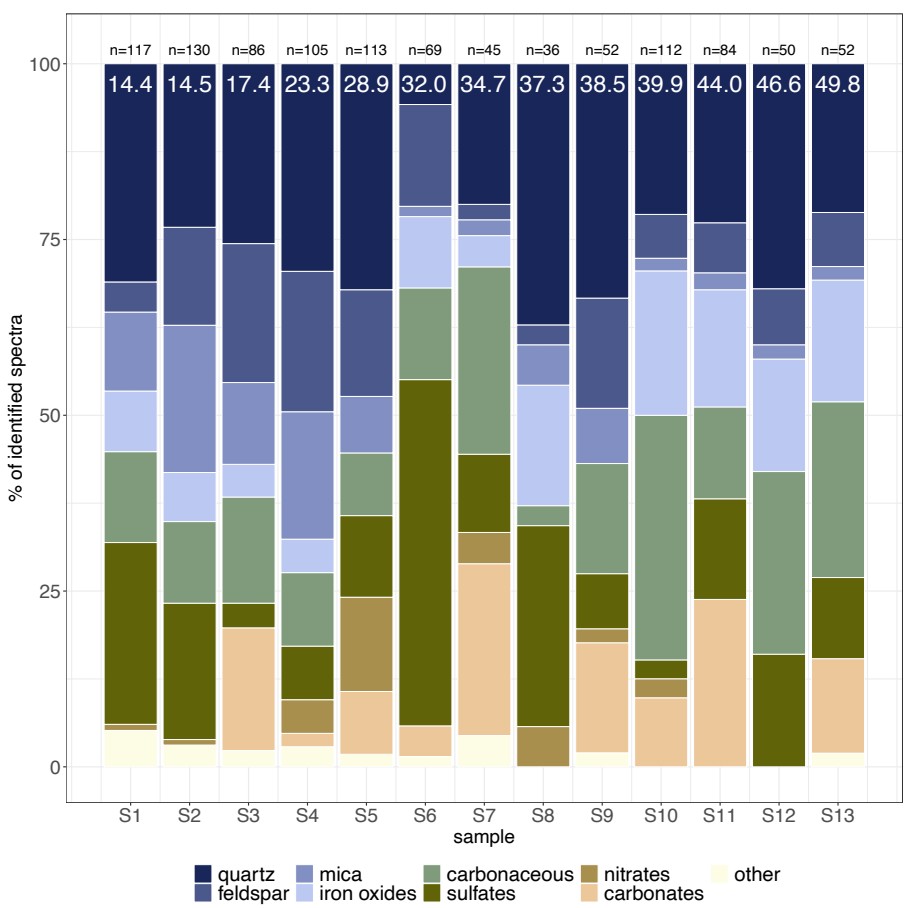

**Figure 3.** Identified Raman spectra of micro-inclusions in EGRIP glacial ice; n is the total number of identified spectra per sample. Age shown in white in ka before 2000 CE (Gerber et al., 2021). For better visibility, some Raman spectra are condensed into groups. Iron oxides are hematite and magnetite, carbonates are dolomite and calcite, and sulfates include gypsum, bloedite, hexahydrite, Na and/or Mg- and undefined sulfates. Other includes rutile, titanite, anatase, epidote, whitlockite, grossular, datolite, and pumpellyite.

### 3.3.2 Mineralogy throughout the last glacial

We found quartz, carbonaceous particles, feldspar, gypsum, and mica at every depth (Fig. 3). Calcite and the chemically-related mineral dolomite occur in S3, S5, S6, S9, S10, S11, and S13. Only dolomite or calcite was found in S4 and S7, respectively. In S6, we identified a total of 27 non-gypsum sulfate minerals, often located in clusters or lined up, in addition to 7 gypsum inclusions. Carbonaceous particles are the dominant species in S7, S10, and S13. Hematite was found in 11 of 13 samples, K-nitrates in 8 samples, and rutile in 6 samples. The remaining minerals occur at a few specific depths.

In general, there is a decrease in the number of different minerals with depth. Shallow samples (S1-S6) consist of 9 to 14 different minerals while deeper samples (S7-S13) consist of 6 to 10 different minerals per sample.

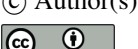



In the shallower region of the core, we identified the rarest minerals (n=<3), such as grossular (S1), anatase (S1, S4), epidote

(S1, S5), pumpellyite (S2), and whitlockite (S2). Titanite (n=3) is the only rare mineral, which occurs in samples throughout the glacial (S4, S9, and S13).

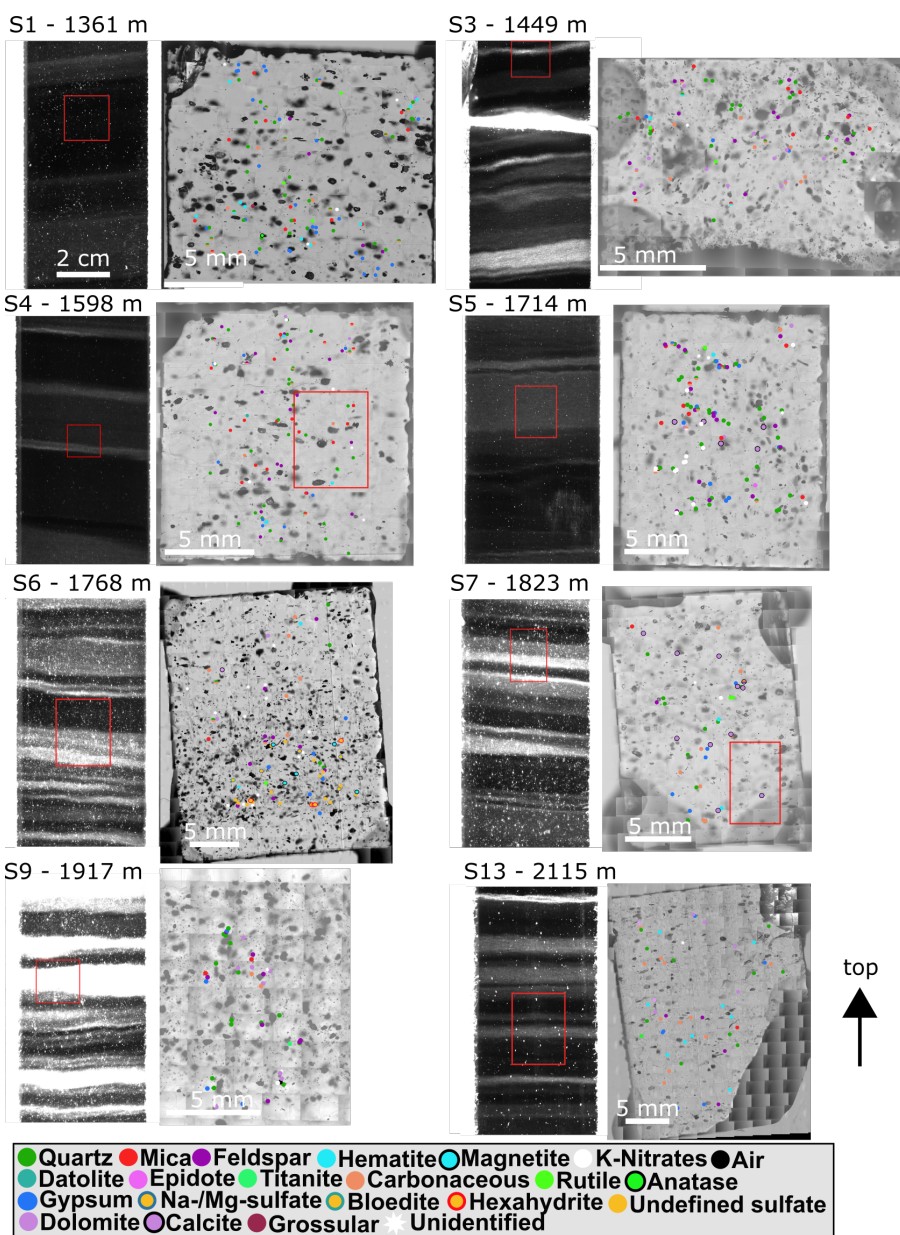

**Figure 4.** Visual stratigraphy (left) and impurity maps from Raman spectroscopy (right), red rectangles are the area of Raman analysis. Locations of identified micro-inclusions are indicated by filled circles. Red rectangles in S4 and S7 indicate areas of LA-ICP-MS 2D imaging displayed in Fig. A3.





### 3.3.3 Localisation of micro-inclusions

Visual inspection shows that most micro-inclusions are in the grain interior. Gypsum, and other sulfate minerals in S6, are often located in dense clusters or lined up behind each other. Interestingly, micro-inclusions showing more than one Raman spectra
were most often associated with carbonaceous particles indicating a tight clustering or merging of carbonaceous particles with other minerals. Cloudy bands display a much higher concentration of visible micro-inclusions, while the surrounding darker areas contain comparably few micro-inclusions.

### 3.3.4 Mineralogy in cloudy bands

The mineralogy in and surrounding the pronounced cloudy bands in samples S6-S11 (Fig. 4, 5, 6, and A1) displays some
distinct features. Some minerals, such as hematite and carbonaceous particles, show a distinct localisation inside cloudy bands. S10 is a prime example of this, characterised by a cloudy band in the upper part mainly containing hematite, and a thicker cloudy band mainly containing carbonaceous particles in the bottom part of the sample (Fig. 5). The various cloudy bands in S11 (Fig. 6) make it difficult to clearly distinguish between them, but a weak mineral localisation also occurs in this sample. In contrast, the thick cloudy band in S9 consists of different minerals without preferred locations.
Even in samples with faint cloudy bands (e.g., S3 and S13), minerals tend to be localised in specific regions. Carbonates and nitrates are also strongly localised in some samples (S3, S4, S5) but less pronounced than e.g., hematite. Common dust minerals, such as quartz, mica, and feldspar, are found throughout the samples, but occur more regularly inside cloudy bands. Gypsum is also common throughout entire samples and tends to cluster but shows no distinct relation to cloudy bands.

### 3.4 2D impurity imaging using LA-ICP-MS

In general, there are strong differences in the microstructural localisation of the analysed elements (Fig. 5, 6). Soluble Na is usually located at grain boundaries. However, especially in the samples containing strong cloudy bands, Na can also be located in the grain interior on some occasions. Mg is predominantly located in the grain boundaries but also occurs infrequently in the grain interior. Al is the prime indicator of insoluble particles (Bohleber et al., 2022) and is found commonly in the grain interior where it often accumulates in particle clusters, i.e. containing several pixels, with additional presence of Ti and Fe.
The latter two also show a comparatively weaker presence at grain boundaries, potentially indicating a soluble component. It is important to note that mass 48 ($^{48}$Ti) can also contain a small contribution of Ca, which would also have a substantial soluble component.

Constituting the most important finding, cloudy bands can be clearly distinguished in the chemical images through the presence of insoluble particles mostly indicated by Al, Ti and Fe at locations that are consistent with the findings from cryo-Raman
analysis (S2 and S10 in Fig. 5, 6). Samples with distinct cloudy bands, i.e. S8 and S10, also show a higher amount of impurities in the grain interiors than shallower samples with less distinct cloudy bands, i. e. S2 and S4. The images generally show a high degree of heterogeneity in elemental ratios indicating a non-homogeneous composition of particle clusters, analogue to the findings by Bohleber et al. (2022).





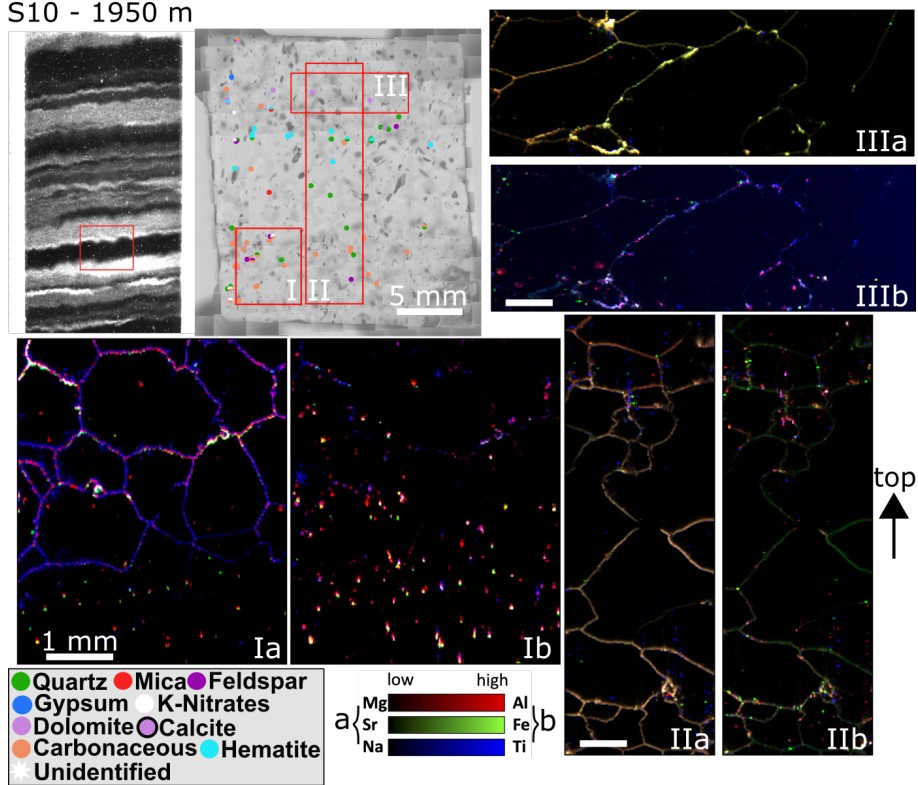

**Figure 5.** Visual stratigraphy (upper left), Raman spectroscopy (upper centre), and LA-ICP-MS 2D imaging of S10, red rectangles are the area of further analyses. Locations of micro-inclusions identified via Raman spectroscopy are indicated by filled circles. LA-ICP-MS impurity images of the same sample in 20 $\mu$m resolution for Mg, Sr, Na and Al, Fe, and Ti indicated by red rectangles and roman numbers in the Raman impurity map, scale is always 1 mm. Measurements were performed on different days, the analysed surfaces thus vary.

## 4 Discussion

### 4.1 Grain size evolution

Mean grain sizes in the glacial are relatively constant with depth until 1700 m except for the larger grains at 1600 m (Fig. 1). This evolution may be due to subgrain formation resulting in smaller grain sizes than in shallower ice (e.g., Duval and Castelnau, 1995; Faria et al., 2014). The steady increase between 1700 and 2121 m is probably due to grain growth and dynamic recrystallisation. Furthermore, the spread in mean grain size is largest in the deepest part of the analysed area. The presented data enable the most detailed insights into grain size evolution in an ice core. An in-depth investigation of the grain size evolution on the centimetre scale would be beneficial but is beyond the scope of this study.

The EGRIP grain size evolution in the glacial ice is similar to the evolution of the NEEM ice core (Montagnat et al., 2014) (Fig. 7). Grain size in the EGRIP ice core develops slightly faster with depth possibly due to the different ice core locations



**Figure 6.** Visual stratigraphy (left), Raman spectroscopy (centre), and LA-ICP-MS 2D imaging (right) of S2, S4, S8, and S11, red rectangles are the area of further analyses. Locations of micro-inclusions identified via Raman spectroscopy are indicated by filled circles. LA-ICP-MS impurity images of the same samples in 20 $\mu$m resolution for Mg, Sr, Na and Al, Fe, and Ti indicated by red rectangles in the Raman impurity maps.



and the impact of the higher dynamics in NEGIS. However, the grain size evolution within both cores is still comparable and
the fast-flowing ice within NEGIS does not (yet) intensely affect the grain size in the upper 2121 m (or upper 80% of ice sheet
thickness).

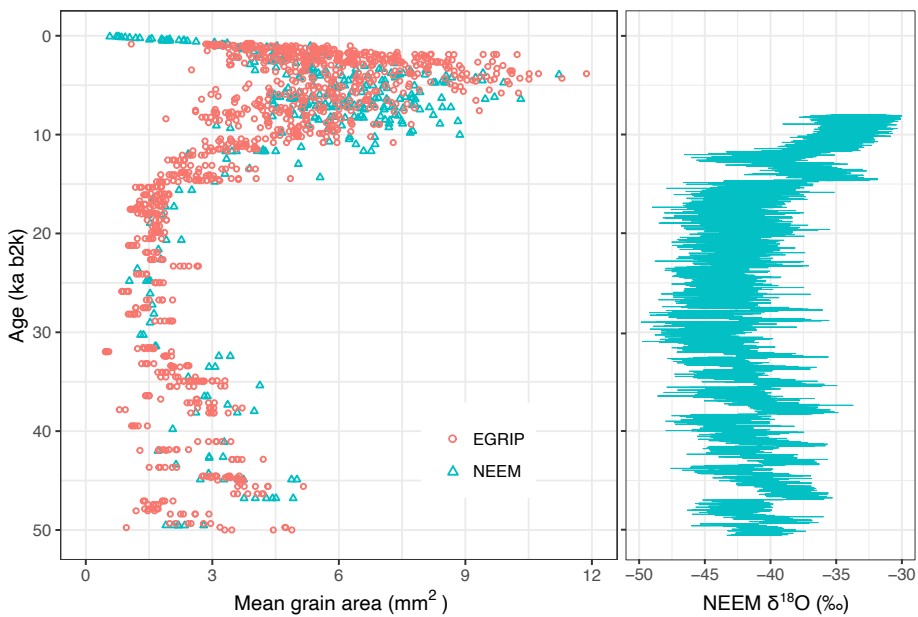

**Figure 7.** Grain size evolution in the EGRIP (data until 1360 m from Stoll et al. (2021a)) and NEEM (Montagnat et al., 2014) ice cores in
relation to stable water isotope data. Age for EGRIP and NEEM from Mojtabavi et al. (2020); Gerber et al. (2021) and Rasmussen et al.
(2013), respectively. NEEM stable water isotope $\delta^{18}O$ record from Gkinis et al. (2021).

## 4.2 Mineralogy derived via Raman spectroscopy

The presented data enable a detailed systematic overview of micro-inclusions over the, to date, entire EGRIP ice core extended
by LA-ICP-MS 2D impurity imaging expanding the recent observations of micro-inclusions in the EGRIP ice core (Stoll et al.,
2021a, 2022).

### 4.2.1 Mineral diversity

Micro-inclusions in EGRIP glacial ice are mainly terrestrial dust minerals, such as quartz, carbonaceous particles, feldspar,
mica, and hematite. Gypsum is another abundant mineral and the only sulfate found throughout the entire core, other sulfate
minerals were only found in S6. In general, the mineralogy is slightly less diverse than in the Holocene (Stoll et al., 2022)
mainly due to a comparable rich diversity of sulfate minerals observed in the top 900 m. However, mineral diversity in the
glacial is comparably constant and implies a more differentiated trend between the upper 900 m and the rest of the core.





The highest number of unidentified minerals (n=8) occurs in S6 indicating that the total amount of different minerals at this depth is higher than identified. Together with data from Stoll et al. (2022) this draws an image of decreasing mineral diversity with depth (Fig. 8), peaking in the intermediate Holocene and remaining relatively constant throughout the last glacial. This

finding is thought-provoking, because the glacial is characterised by a higher amount of exposed bedrock, more dust in the atmosphere, and stronger storms theoretically enabling a higher and possibly more diverse input of aerosols, which is not the case in the Holocene. Especially samples from intermediate depths do not show a high mineral diversity (Fig. 8).

### 4.2.2 Mineralogy and possible inclusion origins throughout 2120 m of EGRIP ice

We show the dominance of terrestrial dust minerals in cloudy bands accompanied by gypsum, carbonaceous particles, calcite,

and nitrates. Together with results from the upper 1340 m of the EGRIP core (Stoll et al., 2022), a detailed picture of the mineralogy and its evolution with depth emerges. We show the 9 most abundant mineral groups in Fig. 8. Sample numbers in this section thus refer to the sample numbers used in Fig. 8 and include results from Stoll et al. (2022). These data enable us to discuss possible source rocks of the observed minerals and geochemical reactions taking place in ice; discussing source regions is beyond the scope of this study.

The first row in Fig. 8 represents minerals from sedimentary carbonate rocks and traces of wildfire, the second one minerals from igneous or metamorphic rocks, and the third one trace minerals and minerals possibly (partly) formed in the ice by chemical reactions.

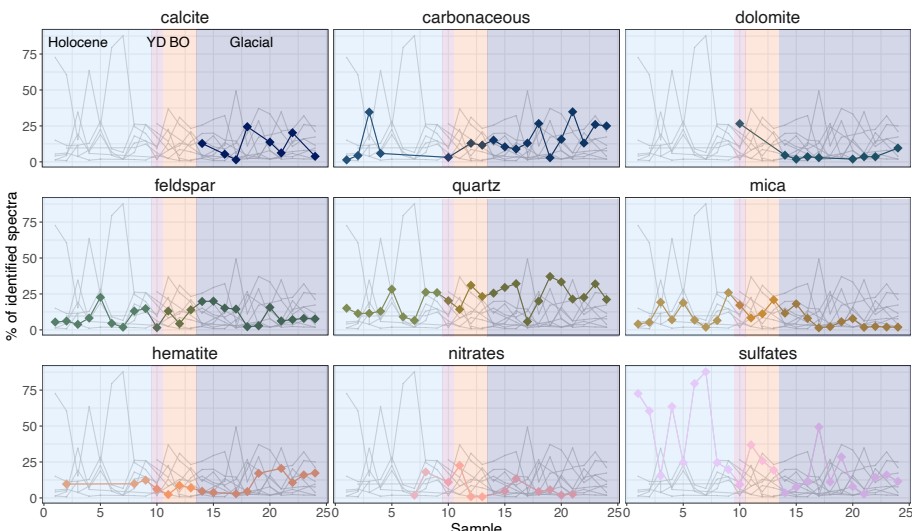

**Figure 8.** Frequently observed minerals (at least once above 20% relative share) throughout 24 samples within the EGRIP ice core (shallowest 11 samples from Stoll et al. (2022)). Samples range from 138 m and 1.0 ka (sample 1) to 2115 m and 49.8 ka (sample 24) in depth and age, respectively. Holocene, Younger Dryas (YD), Bølling–Allerød (BO), and the last glacial are indicated by different shadings.





Dolomite and calcite (first row in Fig. 8) imply that their source rocks are sedimentary carbonate rocks, such as limestone or dolostone. These rocks form by e.g., fossil accumulation and organism activity and involve water and dissolved carbon-
ates. Dolomite and calcite only occur at and below intermediate depths and for the first time in sample 10 and sample 14, respectively. Dolomite occurs abundantly in sample 10 from the dust-rich Younger Dryas and in lower numbers in the deepest samples. Calcite and dolomite are usually found in the same samples indicating that they are transported together and might originate from similar source regions bearing carbon repositories. Interestingly, we did not find calcite and dolomite in the Bølling–Allerød (Fig. 8).

Carbonaceous inclusions likely originate from wildfires (black carbon) or are pure graphite. They are more abundant in the glacial than in the EGRIP Holocene where carbonaceous particles were only found in 5 out of 11 samples and were never the dominant species (Stoll et al., 2022). This regular, and comparably high, abundance is due to higher wildfire activity and higher dust loads in the glacial (Han et al., 2020). Our observation of the mixing of carbonaceous particles with other minerals was partly also observed by Eichler et al. (2019), which identified carbon-sulfate mix particles.

Terrestrial dust minerals, such as quartz, feldspar, and mica, occurring together (second row in Fig. 8) could imply a finger-print of their source rocks being of igneous or metamorphic origin, such as granite or gneiss, which mainly consist of these minerals. These rocks are common in the continental crust and available on the surface as the product of weathering (e.g., sand). These dust minerals also display an opposite abundance behaviour to sulfates, especially in the upper 7 samples. Quartz, feldspar, and mica peak in sample 5 correlating with a small relative number of sulfates. Below sample 7, this trend is weaker
but pronounced in samples 14, 17, and 20 (Fig. 8).

Hematite (third row in Fig. 8) origins from weathering in soil, banded iron formations or places with standing water and can occur in low abundances together with minerals from igneous or metamorphic rocks. Hematite occurs regularly throughout the core, following its low abundance in rocks but high chemical stability against weathering, with minimum numbers in sample 11 and comparably high numbers in the deepest part of the core (samples 19-24) (Fig. 8).

Nitrates and sulfates in the third row in Fig. 8 are minerals, which might act as components in the geochemical reactor ice (Eichler et al., 2019; Baccolo et al., 2021). Nitrates usually originate in arid areas as soil components together with sulfates and sand in e.g., caliche. Nitrates can also form by chemical reactions in the atmosphere during transport to the ice sheet (e.g., Mayewski et al., 1993; Röthlisberger et al., 2002; Iizuka et al., 2008). Nitrate minerals are usually highly soluble and might react with strong acids in the ice resulting in the solution of $NO_3$ (Eichler et al., 2019). In ice, $NO_3$ is probably relocated to the
grain boundaries as solid solution or liquid acid. Thus, solid nitrates can form in the ice by the reaction of $NO_3$ with chloride salts if there is little $H_2SO_4$ and carbonate (e.g. Mayewski et al., 1993; Iizuka et al., 2008). 30 inclusions were identified as Nitrates at 8 depths (S1, S2, S4, S5, S7-S10), which is comparable in number to the 39 nitrates observed in 4 samples below 899 m (Stoll et al., 2022). We only found single nitrate particles while Ohno et al. (2005) found compounds containing both nitrates and sulfates.

Sulfate minerals originate in evaporite depositional environments or oxidising zones of sulfide mineral deposits and can be deposited on the ice sheet via dry deposition. The large number of sulfate minerals in the distinct cloudy band at the bottom of S6 (Fig. 4) implies the possibility of dry deposition events unloading sulfate minerals on ice sheets. In ice, sulfate minerals





can also precipitate in solid form instead of liquid $H_2SO_4$ solutions (Iizuka et al., 2008). They are the dominant species in the shallowest 7 samples while they occur less often in deeper samples mainly due to the large variety of different sulfate minerals in the upper 900 m of the EGRIP core (Stoll et al., 2022). Our findings thus support the almost complete absence of other sulfate minerals than gypsum in Greenlandic glacial ice (e.g., Iizuka et al., 2008; Sakurai et al., 2009; Stoll et al., 2022).

Of the minerals, which never had a relative share of above 20 %, only anatase, epidote, magnetite, rutile, and titanite were identified in several samples (Fig. A4). These minerals were probably transported together with more common minerals and are of detrital origin. Especially the minerals observed only at one depth, such as datolite, grossular, prehnite, pumpellyite, pyromorphite, and whitlockite are less abundant in the Earth´s crust than frequently observed minerals. Rutile, anatase, and epidote were each only found once similar to observations by Stoll et al. (2022) in shallower EGRIP ice. Whitlockite, pumpellyite, and grossular were identified for the first time in ice cores. These minerals occur all over the globe but are not typical dust minerals and a high number of analysed inclusions is thus necessary to find them. Pumpellyite occurs as a secondary mineral in altered gabbro and basalt, and in metamorphic schists. Grossular is part of the garnet group and usually occurs in metamorphosed calcareous rocks. Whitlockite is found in phosphate-rock deposits and igneous pegmatites.

### 4.3 Integrating Raman spectroscopy and LA-ICP-MS 2D imaging data

The presented LA-ICP-MS 2D imaging data expand the knowledge about the localisation of the total amount of solid and dissolved impurities in ice. We confirm recent results from Antarctic ice by Bohleber et al. (2020, 2021) regarding the localisation of Na, Mg and Sr widen the analysis to elements such as Al, Ti and Fe, all with connection to insoluble particulate matter. Furthermore, this is the first high-resolution data, i.e. using laser spot sizes of 20 $\mu$m, from cloudy bands enabling more detailed insights than before. The found particle clusters with high intensities of Fe and Ti are in accordance with the identification by Raman spectroscopy of the Fe-bearing minerals magnetite and hematite and the Ti-bearing minerals rutile, anatase and titanite. Localised minerals, such as hematite in cloudy bands, are often found with LA-ICP-MS in clusters of high Fe intensity (Fig. 5, 6). Furthermore, the visible particle aggregates are found dominantly in the grain interior similar to observations from this study and earlier works (Eichler et al., 2017; Stoll et al., 2021b, 2022). LA-ICP-MS data help confirm relatively inconspicuous Raman spectra that are interfered with by the overlying ice spectrum, as in the case of magnetite. It also provides a different perspective on the differentiation in the localisation of insoluble and soluble impurities.

LA-ICP-MS revealed that the impurity intensities of all analysed elements are higher in cloudy bands than in the surrounding areas. Furthermore, insoluble particles are particularly abundant in areas belonging to cloudy bands as predicted by microstructure mapping and Raman spectroscopy. The resolution of 20 $\mu$m enables detailed characterisations of the chemistry of EGRIP glacial ice, especially in samples containing cloudy bands where particle clusters are the main characteristic.

### 4.4 Deciphering the origin of cloudy bands by combining visual and chemical data

Based on our detailed data set of different visual and chemical parameters throughout EGRIP glacial ice we classified cloudy bands into seven different types differing e.g., in thickness and brightness. Complemented by novel insights into their chemistry





we show that cloudy bands are more complex and diverse than so-far known. We thus try to expand the understanding of their origin by discussing processes involved in their formation.

Cloudy bands are commonly interpreted as storm events in spring or summer transporting large amounts of dust from Asian deserts across the Greenland ice sheet (Svensson et al., 2000). If cloudy bands were solely the result of dust deposition events in spring and summer, dark layers would dominate the visual stratigraphy data. However, the line scan images are dominated by

bright layers of high insoluble particle content as displayed by our Raman spectroscopy and LA-ICP-MS data. An explanation for the greater thickness of cloudy bands compared to neighbouring dark layers is that the largest amounts of accumulation occur in Greenland in spring and summer. Nevertheless, this is not sufficient to explain the variations we see in cloudy band thickness.

To understand the anatomy and origin of cloudy bands we must consider the main types of dust deposition: 1) dry deposi-

tion, i.e. fallout of solid impurities over the ice sheet by atmospheric transportation, and 2) wet deposition, i.e. snowfall and accumulation, including washout of atmospheric dust particles.

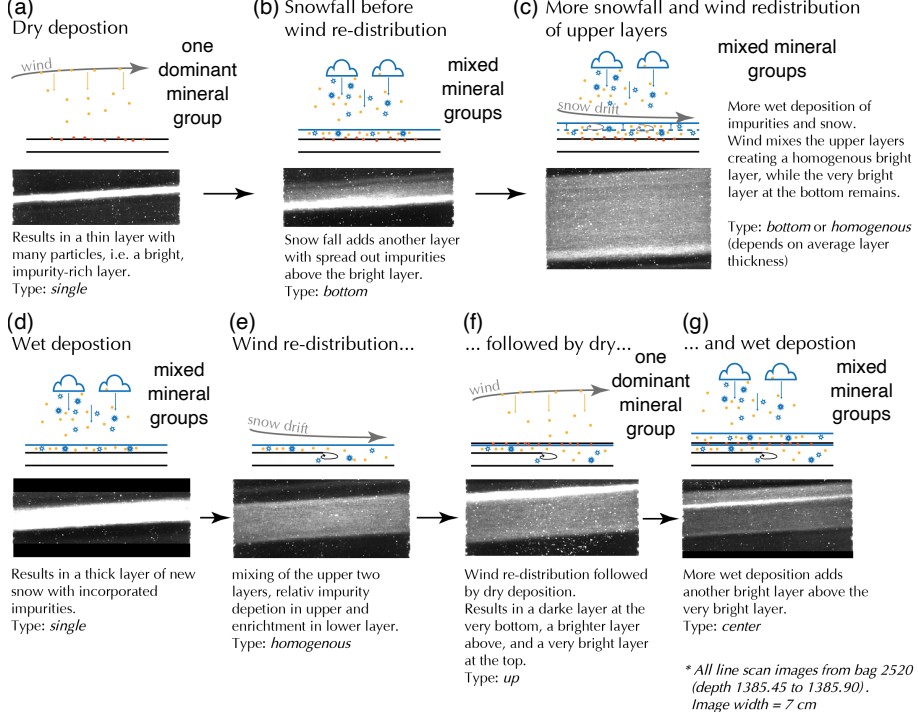

**Figure 9.** Possible processes leading to the identified cloudy band types.

We interpret thin and bright layers in the visual stratigraphy (*single*) as long-lasting dry precipitation events, which are either formed by precipitation or by wind-redistribution of surface snow, similar to Svensson et al. (2005). The authors further suggest that very thin and bright cloudy bands are associated with enhanced scavenging early in a snowfall event or as dry

deposition. A single thin and bright layer making up a cloudy band or part of it indicates dry deposition (Fig. 9 a), as the layer





thickness does not increase by accumulation but many dust particles are deposited on the same surface. The greater the number of deposited particles in a single layer, i.e. without a precipitation event in between, the brighter it will appear in the visual stratigraphy. Some cloudy bands show a distinct layering of specific minerals, such as hematite and carbonaceous particles, and particles of insoluble chemistry (Al, Fe, Ti) (S10 in Fig. 5). If wind speeds are low, i.e. no significant redistribution of
surface snow, the thin dry deposition layer can be covered by wet precipitation adding new snow (Fig. 9 b). Wet deposition would lead to an increasing layer thickness during precipitation creating not a thin cloudy band, but one of greater thickness as wet means: a layer of impurities and snow (Fig. 9 c). Impurities would be more or less evenly distributed (e.g., S1, S5, S11, S12 in Fig. 4, 6) in this layer of newly fallen snow, as opposed to dry deposition where little snow is involved and thus most impurities are concentrated in one thin layer (S10 in Fig. 5). Furthermore, snowfall events are short-lasting, i.e. a few days,
rather than months, and the sequence of snowfall events would thus create a more alternating pattern of bright and darker layers in the stratigraphy (Fig. 9 g). Wet deposition events are thus unlikely to create homogeneous cloudy bands. The thin and bright cloudy band can get mixed with a dark layer from below by surface processes and result in a medium bright thick layer (Fig. 9 d and e).

Post-depositional processes, such as redistribution of surface snow by the wind, would lead to a well-mixed surface snow
layer (Amory, 2020). This layer then contains a mix of the previously deposited impurities (e.g., S5, S7 and S9 in Fig. 4 ) and would appear as a homogeneous and moderately bright layer in the visual stratigraphy data. Snow layers impacted by wind-redistribution processes (Fig. 9 e) could lead to the homogeneous type and to *up*, *centre*, *bottom*, and *confined* all containing a homogeneous layer. Further, wind dunes creating high-density layers in the snow could also serve as lids (Birnbaum et al., 2010) isolating high impurity layers later visible as a distinct type of cloudy bands (Fig. 9 f), but data from Greenland on these
layers is rare. The redistribution and mixing of surface snow by drifting (<2 m above ground level) and blowing (>2m above ground level) (Amory, 2020) seem to be the main parameters in creating the observed visual features of the seven distinguished cloudy band types.

Redistribution of surface snow does not disturb methods that analyse variations between seasonal signals in ice cores, as only the seasonal snow is mixed. Mixing snow between different years seems unfeasible, as, even in glacial times, accumulation in
Greenland is high enough for layers to become covered and preserved quickly. Yet this mixing affects the detailed analysis of small-scale distributions of insoluble and soluble impurities. When analysing a single thin cloudy band, it contains the insoluble particles from dry deposition, maybe including some precipitation, but not significant redistribution by surface processes such as wind. The thick homogeneous cloudy bands are usually dimmer than the single thin cloudy bands because impurities are less concentrated. They are thus the result of the redistribution of surface snow. Dark layers seem to be untouched by mixing
with spring and summer layers, as they do not contain many visible impurities. They are thus the effect of fall and/or winter precipitation on the ice sheet. Deciphering mixing rates and other involved processes is beyond the scope of this study.

Single bands mostly appear in combination with other, less bright, cloudy bands (*up*, *bottom*, *centre*, *confined*, or *hetero-geneous*). Assuming that a stack of bright layers between two dark layers, i.e. our definition of a cloudy band, represents one year, then the most common case is one bright layer per year (*single*, *up*, *bottom*, and *centre*). In some cases, we find multiple
thin and bright cloudy bands within the stratigraphy of one year (*confined* and *heterogeneous*). In other cases, the thin cloudy





band is entirely missing (*homogeneous*). Assuming a constant influx of dust particles over the Greenland ice sheet on an annual scale with seasonal variations, the absence of a single thin cloudy band can then be used as a proxy for a year with significant surface snow redistribution processes. The intensity of cloudy bands thus indicates the sequence of events taking place on the ice sheet over time.

### 4.5 Outlook

This study on cloudy bands shows the value and future potential of a multi-method and multi-scale approach. To expand our explanation of the development of cloudy bands, a similar study on Antarctic ice is needed. However, continuous visual stratigraphy data is rare and, to our knowledge, only available for the EDML ice core (Faria et al., 2018). Another valuable approach is comparing our results with cloudy bands in NEEM, NGRIP, and RECAP ice. Furthermore, more information on the impact of inclusions on the brightness in the visual stratigraphy is needed. What impact do particle grain size, shape, and inclusion-chemistry have on the observed brightness? Is it possible to derive more information on impurities from visual stratigraphy? Future LA-ICP-MS applications could be: 1) analysing larger areas (several centimetres) covering entire cloudy bands and their surroundings, and 2) higher resolution (1-5 $\mu$m) to distinguish analytes in the observed impurity clusters more clearly. At last, correlating data on cloudy band types and chemistry with millimetre-scale grain size and shape evolution, borehole deformation, and c-axes orientation data would enlighten internal deformation within the EGRIP ice core.

### 5 Conclusions

Cloudy bands are the main visible feature in glacial ice from deep polar ice cores. They are important, but poorly understood, factors regarding climatic reconstruction and the internal deformation of ice. With this study, we conducted the first systematic analysis of cloudy bands in general, and in detail, on ice from the East Greenland Ice-core Project ice core accompanied by an analysis of the grain size evolution. We combine visual techniques such as visual stratigraphy, fabric analyser, microstructure mapping, and chemical methods such as Raman spectroscopy and laser ablation inductively coupled plasma mass spectrometry to bridge different spatial scales resulting in new insights into glacial ice, and especially cloudy bands. We identified seven categories of cloudy bands. Single cloudy bands are by far the most abundant ones. However, the relative abundance of cloudy band types differs with depth and the prevailing period (Stadial or Interstadial). The main minerals in EGRIP glacial ice are quartz, mica, feldspar, gypsum, and carbonaceous particles, which we identified at every depth. The mineralogy is slightly less diverse than in EGRIP Holocene ice. Some cloudy bands show a dominant mineral species (e.g., hematite or carbonaceous particles) indicating a strong deposition event that is preserved with depth. Laser ablation inductively coupled plasma mass spectrometry 2D imaging shows that cloudy bands are distinguishable from the surrounding ice and bulk results agree well with other methods. Mostly dissolved analytes, such as Na, are mainly at the grain boundaries; insoluble analytes, such as Fe and Al, are arranged in particle clusters similar to Raman spectroscopy observations. Finally, we elaborate on theories about the origin of cloudy bands based on our chemical and visual observations, thus laying the foundation for future work tackling their direct impact on deformation.





*Data availability.* All data sets from this study will be available online at PANGAEA. Visual stratigraphy data is available at Weikusat, I et al. (2020); https://doi.org/10.1594/PANGAEA.925014.





## 5.1 Appendix A1

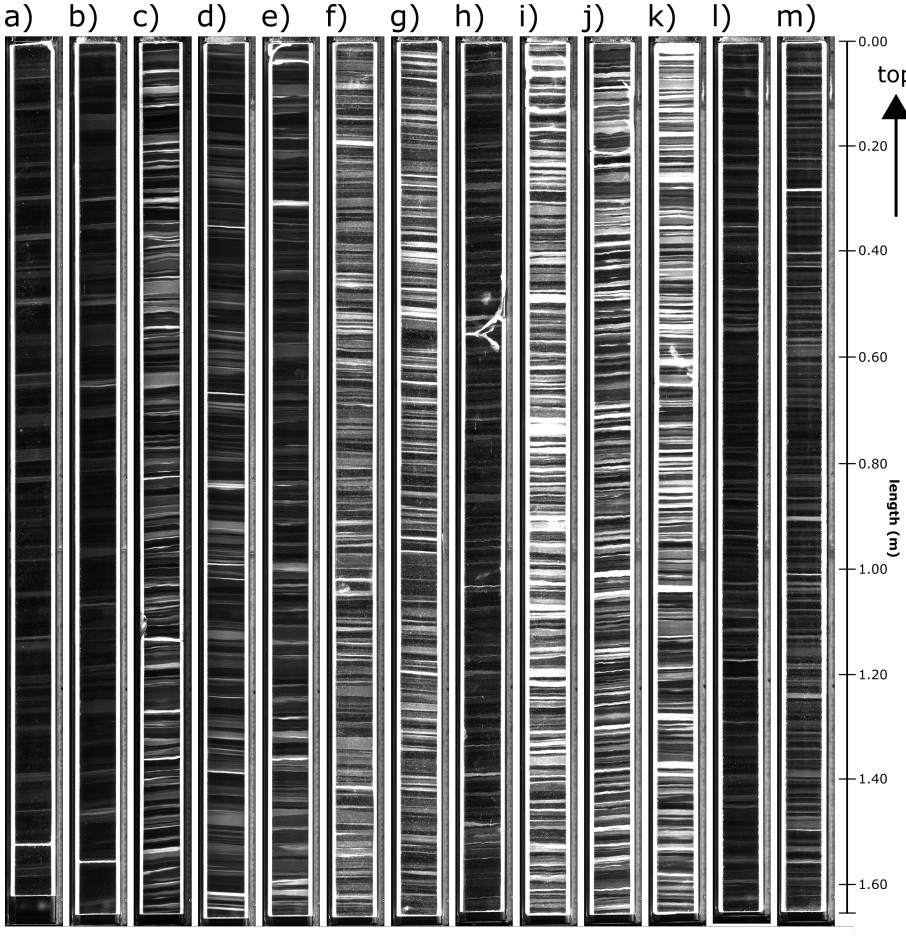

**Figure A1.** Visual stratigraphy images of the 13 ice core samples partly analysed with Raman spectroscopy. Each sample is 1.65 m long and 8-9 cm wide. Scans consist of three images with different focus planes and apertures. The samples cover the depth regime between 1360 and 2115 m and an age regime between 14.4 and 49.8 ka, i.e. the last glacial (Gerber et al., 2021).





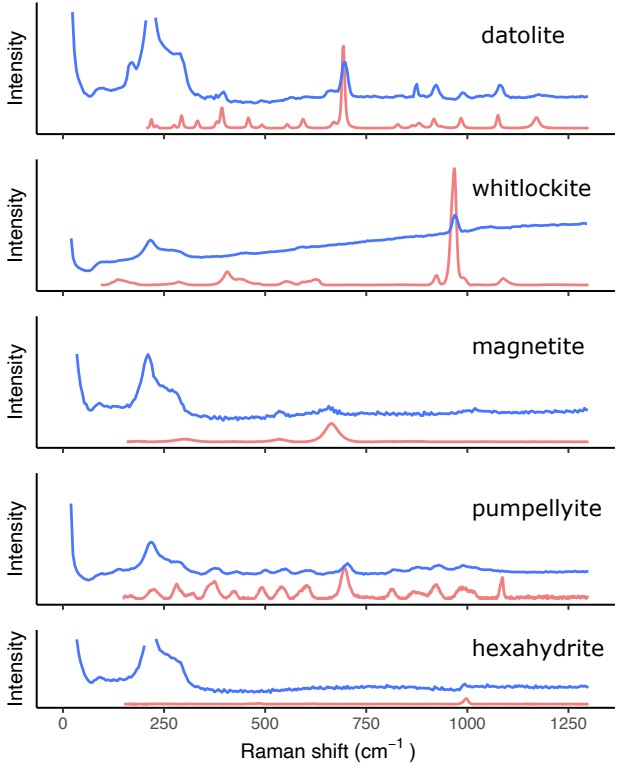

**Figure A2.** Measured spectra (blue) of datolite, whitlockite, magnetite, pumpellyite, and hexahydrite compared to reference spectra (red) from the RRUFF database (Lafuente et al., 2015). Small deviations are due to the overlaying ice spectrum and differences in the used devices.



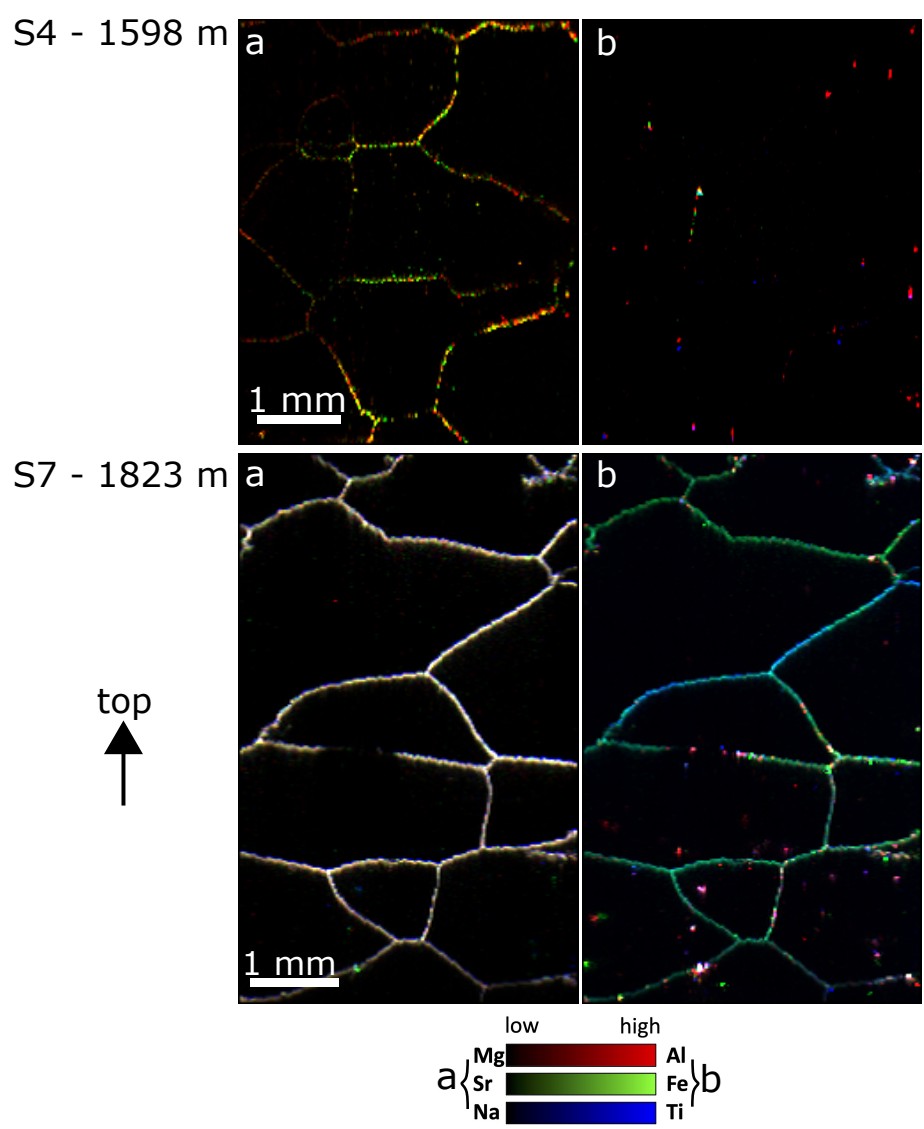

**Figure A3.** LA-ICP-MS 2D impurity images of S4 and S7 in 20 $\mu$m resolution for Mg, Sr, Na and Al, Fe, and Ti.

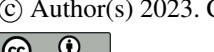



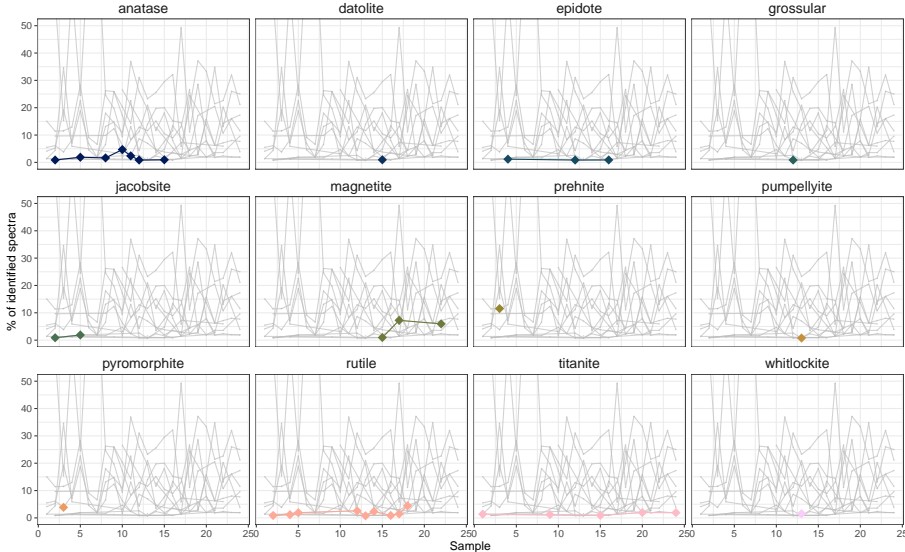

**Figure A4.** Sparsely observed minerals (always below 20% relative share) throughout 24 samples within the EGRIP ice core. Data from the first 11 samples from the Holocene from Stoll et al. (2022). Note the changes on the y-axes compared to Fig. 8.



## Appendix A3

**Table A1.** Identified Raman spectra in EGRIP glacial ice.

| Mineral | Number | Formula |
|---|---|---|
| Quartz | 268 | $SiO_2$ |
| Carbonaceous | 170 | C |
| Gypsum | 134 | $CaSO_4 * 2H_2O$ |
| Feldspar | 119 | $(K/Na/Ca/NH_4)(Al/Si)_4O_8$ |
| Mica | 92 | $(K/Na/Ca/NH_4)Al_2(Si_3Al)O_{10}(OH)_2$ |
| Hematite | 86 | $Fe_2O_3$ |
| Calcite | 62 | $CaCO_3$ |
| K-Nitrates | 30 | $KNO_3$ |
| Dolomite | 25 | $CaMg(CO_3)_2$ |
| Sulfate undefined | 15 | $XSO_4$ |
| Magnetite | 11 | $Fe_3O_4$ |
| Rutile | 10 | $TiO_2$ |
| Hexahydrite | 7 | $MgSO_4 * 6H_2O$ |
| Air | 5 | $O_2$ |
| Na and/or Mg-sulfate | 4 | $NaSO_4$ and/or $MgSO_4$ |
| Titanite | 3 | $CaTiSiO_5$ |
| Anatase | 2 | $TiO_2$ |
| Epidote | 2 | $Ca_2(Fe/Al)Al_2(Si_2O_7)(SiO_4)O(OH)$ |
| Whitlockite | 2 | $Ca_9Mg(PO_4)_6(PO_3OH)$ |
| Bloedite | 1 | $Na_2Mg(SO_4)_2 * 4H_2O$ |
| Datolite | 1 | $CaB(SiO_4)(OH)$ |
| Grossular | 1 | $Ca_3Al_2(SiO_4)_3$ |
| Pumpellyite | 1 | $Ca_2(Mg/Fe/Al/Mn)Al_2[Si_2O_6(OH)][SiO_4](OH)_2OH/O$ |
| Total | 1051 | |



*Author contributions.* Initial manuscript idea by NS and JW. NS performed microstructure mapping and Raman spectroscopy analyses and data processing and analysis. JW, NS, and AS performed visual stratigraphy measurements; visual stratigraphy data were analysed by JW. PB and NS acquired and analysed LA-IPC-MS data. The manuscript was written by NS, JW, and PB with the assistance of all co-authors.

*Competing interests.* The contact author has declared that neither they nor their co-authors have any competing interests.

*Acknowledgements.* This work was carried out as part of the Helmholtz Junior Research group "The effect of deformation mechanisms for ice sheet dynamics" (VH-NG-802). Nicolas Stoll thankfully acknowledges additional funding from the graduate school POLMAR. We especially thank the EGRIP physical properties team, for example, Jan Eichler, Johanna Kerch, Ina Kleitz, Daniela Jansen, Sebastian Hellmann, Wataru Shigeyama, Ernst-Jan Kuiper, Tomoyuki Homma, Steven Franke, and David Wallis. We thank all EGRIP participants for logistical
support, ice processing, and fruitful discussions. EGRIP is directed and organized by the Centre for Ice and Climate at the Niels Bohr Institute, University of Copenhagen. It is supported by funding agencies and institutions in Denmark (A. P. Møller Foundation, University of Copenhagen), USA (US National Science Foundation, Office of Polar Programs), Germany (Alfred Wegener Institute, Helmholtz Centre for Polar and Marine Research), Japan (National Institute of Polar Research and Arctic Challenge for Sustainability), Norway (University of Bergen and Trond Mohn Foundation), Switzerland (Swiss National Science Foundation), France (French Polar Institute Paul-Emile Victor,
Institute for Geosciences and Environmental research), Canada (University of Manitoba) and China (Chinese Academy of Sciences and Beijing Normal University). Pascal Bohleber gratefully acknowledges funding from the European Union's Horizon 2020 research and innovation program under the Marie Skłodowska-Curie grant agreement no. 101018266. Julien Westhoff and Dorthe Dahl-Jensen thank the Villum Foundation, as this work was supported by the Villum Investigator Project IceFlow (no. 16572).



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
