# Peer review of "Chemical and visual characterisation of EGRIP glacial ice and cloudy bands within"

_The Cryosphere, 2022_

## Author Comment (AC1)

**Comment on tc-2022-250**

Giovanni Baccolo (Referee) **(R1)**

**R1:** This is an extremely interesting contribution about the analysis of cloudy bands in glacial ice found in the EGRIP ice core. The authors applied a set of cutting edge techniques, providing valuable data about the micro-structure and –composition of these well-known features that are found in polar ice from the glacial periods. Despite most ice core scientists that have worked with polar ice cores know what a cloudy band is, most of them does not really know what is behind their formation and significance. Considering this, this work is more than welcome and will represent a benchmark for future studies dealing with the structure and description of glacial ice in polar ice cores.

The manuscript generally reads well, despite I think that most sections could be shortened. Reducing the text would surely improve the readability of the entire work. The amount of data presented is really impressive: visual stratigraphy, texture, impurity distribution and mineralogy. The integration of all these data is for me the most lacking aspect of the manuscript. Sometimes you have the impression that the authors are just presenting the data, without really discussing them in relation to ice micro-structure or to the paleoclimatic significance of the proxies they are focusing on. Also the cited literature reflects this, there are some discussions or interpretations that do not really take into account what was done in previous study and this of course weakens the discussion of your data. This is probably related to the fact that there is so much material in this manuscript that it is not easy to elaborate it, but I think that more efforts toward this direction would improve the soundness of the study and increase its future impact. My impression are also supported by the fact that the authors say many times that some of the data will not be discussed because such discussions would be out of the scope of the present manuscript. I personally don't like finding 3 or 4 times such a statement and suggest to remove them and to consider to reduce all the discussion that is not really linked to the story of this research. In the light of this, I found that after some revision the manuscript will be ready for publication in The Cryosphere. Below my more specific comment.

**A:** We thank the referee for his detailed feedback, which certainly improves the manuscript. This study was set-up to be a first benchmark regarding the first integration of three methods, cloudy bands, and EGRIP glacial ice with hopefully some follow-ups as more data becomes available. We condensed the manuscript and discuss the different data sets better while incorporating the suggested changes.

**R1:** Line 2: "concentrations, are"; I would not say "the last glacial", of course your are thinking of Greenland ice cores, where talking about glacial ice means talking about ice from the last glacial period, but for Antarctica this is not automatic, I would rather say "from glacial periods".
**A:** We agree and change it to "glacial periods".

**R1:** Line11: why you say "Cloudy bands are thus clearly distinguishable in the chemical data."? I mean, using thus it seems that in the previous passage, you provided evidences to support what you are stating here, but it does not seem the case. Maybe I have not understood, but I suggest you to rephrase this part of the abstract.
**R1:** We change the sentences and state the mentioned fact earlier. Changed to "2D impurity imaging with 20 mu resolution revealed that cloudy bands are clearly distinguishable in the chemical data. Na, Mg and Sr are mainly at grain boundaries.".

**R1:** Line 15: I would add "in deep polar ice cores" at the end of the passage.
**A:** Changed to "in deep polar ice cores".

**R1:** Line 19-22: I suggest removing the cited drilling sites, this is because, especially for Antarctica, there are so many deep sites that it is impossible to cite all the relevant ones. For example now you are not considering Vostok that is actually the site where the deepest ice core ever has been drilled, or also Dome A, where the future longest ice core is expected. I would simply remove the brackets where you list a few sites, it is impossible to cite them all or to make a rigorous selection. You could also think of citing a review paper about ice core science at this point, for example Brook & Buizert, (2018) or Jouzel (2013) or also Langway (2008) about early polar ice cores drilled in Greenland.
**A:** We removed the examples and refer to Jouzel (2013) and Brook & Buizert (2018) for an overview. Changed to: "Over the last decades, a variety of locations in Greenland and Antarctica were chosen for drilling operations (for an overview see Jouzel, 2013; Brook & Buizert, 2018)."

**R1:** Line 22-23: I suggest changing to "Considering different polar ice cores most of the physical and chemical properties of ice and of its impurities vary, depending on several parameters that are different at each drilling site. Concurrently, there are some features that seem recurrent, such as the presence of the so-called "cloudy bands" in glacial ice."

**A:** We agree and change the sentence to: "Considering different polar ice cores, most of the physical and chemical properties of ice and its impurities vary, depending on several parameters that are different at each drilling site. Concurrently some features seem recurrent, such as the presence of the so-called "cloudy bands" in glacial ice."

**R1:** Line 25 -26: I suggest starting this passage describing how they look visually, so that the reader can understand their name, and then you can describe more in detail their physical structure. I would not say small crystals, rather "crystals smaller (maybe add a reference value here to have an idea of what you mean with small) than the surrounding ice"

**A:** We changed the passage accordingly and start with a visual description. However, adding quantitative grain size values is close to impossible due to the high variability within these bands, and the low amount of published grain size data from cloudy bands (which partly motivated this study). Gow & Williamson (1976) report that "Most of the bands examined are so fine grained (mean crystal diameters on the order of 1.0 mm are not uncommon) that the conventional universal stage and microscope were needed to measure c-axis orientations.", but we refrain from putting this as an absolute grain size description. A detailed grain size and fabric analysis of the cloudy bands in this study was started, but is difficult to conduct due to statistical reasons caused by a comparable small number of grains.

We changed the paragraph to:" Cloudy bands are horizontal, grayish-white stratigraphic layers with thicknesses between 1 mm and several centimetres (Fig. A1) (Gow and Williamson, 1976; Faria et al., 2010). They are characterised by a much finer grain size (~1 mm) than the surrounding ice and contain a very high concentration of micro-inclusions and other impurities (e.g., Ram and Koenig, 1997; Barnes et al., 2002; Svensson et al., 2005; Faria et al., 2010; Eichler et al., 2017). Gow and Williamson (1971, 1976) were among the first to describe cloudy bands in the Byrd ice core, where they observed dirt bands and the much more abundant cloudy bands. Dirt bands contained large particles, detectable by the eye, and were classified as volcanic ash bands. Cloudy bands, however, were not composed of visible debris but of a greyish-white appearance, hence the name (Gow and Williamson, 1971, 1976). The preferred crystal orientation within these bands is clustered about the vertical indicating strong horizontal shearing. Cloudy bands were thus associated with dust and deformation and provisionally interpreted as shear bands (Gow and Williamson, 1971, 1976)".

**R1:** Line 31: I suggest specifying here that grain refers to insoluble particle size, while ice grain are referred across the entire manuscript as ice crystals. This would help not to confuse the reader.

**A:** We adjusted this paragraph to avoid confusion (answer above) and refer to the size of ice crystals consistently as grain size and now clarify this in the text. Insoluble particle size is not discussed in this manuscript due to (unfortunately) no available data.

**R1:** Line34: "There is no typical cloudy band" I don't understand this passage. What do you mean? That until now a unique description for this ice-structure has not yet been established? Or maybe that cloudy bands can significantly differ within each other? Please reformulate

**A:** The statement should summarise that cloudy bands differ in width, brightness, shape (from horizontal to wavy and more), and angle. We rephrase to:" Cloudy bands vary in thickness, brightness, and shape and are thus hard to constrain…".

**R1:** Line 36-39: Do you mean that in some cases the bands correspond to the seasonal peaks of specific analytes? If this is the case this should be better explained.

**A:** Yes, we refer here to e.g. Fig. 5 from Svensson et al. (2005). The visual stratigraphy intensity profile looks similar to many of the impurity records derived by CFA, but not all impurities contribute equally to making cloudy bands visible. For example, sulfate peaks are often not visible in the visual stratigraphy, the same likely apples to nitrate and ammonium peaks. Comparing the grayscale profile with the CFA record could be a dedicated follow up study. Investigating if there are differences in the correlation between both parameters in stadial, interstadial, and Holocene ice would be another interesting approach.

To avoid confusion, we change the text to: "Svensson et al. (2005) show that, in most cases, the brightness variations of visual stratigraphy and cloudy bands match the seasonal cycles of tracers, especially dust, derived by continuous flow analysis (CFA) (Fig. 5 in Svensson et al., 2005)".

**R1:** Line 47-48: suggest to change to "and thus affect the bulk deformation rate of ice softening it"
**A:** Changed as suggested.

**R1:** Line 67: I suggest adding one or two references to papers prepared by the Japanese group that also developed a similar technique. For example (Ohno et al., 2005; Sakurai et al., 2009).
**A:** The sentence refers to works coupling Raman spectroscopy and microstructure mapping to investigate the localization of solid inclusions, this is not done in the mentioned papers. We however added these references as examples for cyro-Raman analyses.

**R1:** Line 68-82: I would change a bit the structure of the introduction here. I would start saying that this work focuses on the study and description of cloudy bands in the EGRIP ice core using the methods that you have just cited. Then I would say that you choose this specific ice core as it is one of better studied from this point of view, so you already know a lot of things about the distribution, quality and quantity of impurities present in the ice and about ice microstructure.
**A:** We changed the structure and rephrased accordingly.

**R1:** Line 90: it is not clear what you mean with grain size. This is probably ice-grain size, but not to confuse the reader I would be consistent throughout the manuscript in distinguishing ice crystals and insoluble impurities using different terms (see my comment at line 31).
**A:** The term grain size for the size of ice crystals has been adapted from geology and is widely used and thus established in "structural glaciology" (see e.g. Montagnat et al., 2014, Faria et al., 2014, Weikuat et al., 2017). Terminology is often an issue in interdisciplinary work, but since there is no particle size data (from e.g. the coulter counter or Abakus) available for EGRIP glacial ice we keep grain size and mention that we always to refer ice crystal size in this study.

**R1:** Line 119: here you say that the bands presenting signs of folding were grouped as "unknown", but a few lines above (line 110) you said that in this study you did not consider deformed cloudy bands. I don't understand.
**A:** This is indeed not very clear. We excluded bags where the majority of cloudy bands were deformed or belong to more than one category. However, if the vast majority of cloudy bands in the bag could be identified and only a small number of cloudy bands were unidentifiable, we classified them as unknown. To clarify this, we add two exampled in the appendix (Fig. A2) and change the text to: "We exclude 55 cm ice cores from the analysis where most cloudy bands show deformation features to eliminate a complication factor. Yet some samples can contain single cloudy bands showing signs of deformation or belonging to more than one group. To also account for these bands, without being able to group them into a certain category, we group them as "unknown" (examples in Fig. A2)."

**R1:** Line 144: I would add "to clean the surface of interest before the analysis"
**A:** Added.

**R1:** Line 145: I would add "to monitor potential instrumental drifts"
**A:** Added.

**R1:** Line 152-153: this passage is not clear, please rephrase.
**A:** Changed to: "The mean grain size increases with depth (Fig. 1), and 9 cm thin sections contain between 471 and 4550 grains in the investigated depth range. Mean grain size decreases from 3.6 $mm^2$ at 1340 m depth to 1-2 $mm^2$ by 1800 m (except 2-2.7 $mm^2$ at 1600 m depth)."

**R1:** Figure1: maybe better saying in the caption that blue bands correspond to the cloudy bands you measured through raman and LA? Because now it seems that you only considered those bands while actually you classified all of them throughout the considered interval depth.
**A:** The blue lines actually represent all 165 cm (3x 55 cm bags) with analysed cloudy bands. Cloudy bands do not occur consistently throughout the last glacial and cannot be continuously analysed for several reasons. This study is merely a first attempt to possibly start a larger investigation of cloudy bands. We mark the samples investigated with Raman spectroscopy now in the mean grain size plot and additionally add some more grain size data points.

**R1:** Line 160: I would remove "However, the relationship between particle size and grayscale is unexplored.", this is not so relevant for your manuscript.
**A:** We do not explore this topic further, but added it to show the value of doing so in future research.

The specific physical interaction between the emitted light from the line scanner and the particles in cloudy bands has, to our knowledge, never been explored but might offer new insights and in the best case, maybe even some sort of proxy. We delete it from the results section, but leave it in the outlook.

**R1:** Figure2: I have noticed that you never discuss probably the most evident feature of this image, that is the frequency increase of unknown cloudy band types at specific depths. Could you add some detail about this in the discussion part of the manuscript? How do you interpret this?
**A:** We briefly mention this in the results section: "The type *unknown* occurs more frequently in deeper ice (Fig. 2a) as layers thin and categorisation becomes more difficult. Furthermore, the darkness of the image plays a role. For consistency, throughout the glacial, the brightness is kept constant for all images, thus making our analysis more favourable for stadials, i.e. cold periods with a higher dust concentration, within the last glacial period." The large number of unknown types from 35 ka is from interstadial ice, where the visual stratigraphy appears much darker (lower dust concentration). We thus add to the discussion: "The abundant unknown cloudy bands at 35, 45, and 48 kyr b2k (Fig. 2) all origin from Interstadials with low dust concentrations hampering their characterisation."

**R1:** Line 180: be careful because you are saying that carbonaceous particles belong to a specific mineral class, but this is not the case
**A:** To avoid confusion we change the sentence to:" The most common mineral is quartz (n=268), followed by not further distinguished particles bearing carbon (from here on carbonaceous particles) (n=170),…".

**R1:** Line 190: this is a curiosity I have. I have seen that you identified hematite rather frequently, what about goethite? From my experience this Fe-oxide is typically more abundant than hematite in polar environments. This is because hematite requires arid environments that are more easy to find in the tropics, while goethite is more related to temperate and subpolar climate (relatively cold and wet conditions). Maybe the fact that you did not identify goethite depends on your technique that did not allow to see it?
**A:** This is indeed an interesting (not-)finding, which has puzzled me as well. As far as I know, no Raman spectroscopy study has identified goethite in polar ice while hematite has been identified by Eichler et al., 2019, and Stoll et al. 2022. The goethite spectrum is partly overlaid by the ice spectrum, but it should still be identifiable. We measured a few spectra with one peak resembling a geothite spectrum peak, however, this is not enough evidence to identify them as such. Furthermore, some iron minerals are easily converted to hematite at laser powers of 1 mW making it impossible to identify the original material and partly if a transformation took place (Hanesch, 2009). However, the same applies to magnetite, which we identified in our samples thus rebutting this possibility. Investigating an ice sample with known goethite content would be the best approach to further investigate this issue.

**R1:** Line227: you say that elemental ratios are highly variable in the particle clusters you identified with la-icpms, but actually you are never showing results about these ratio, is a figure missing?
**A**: We do not show quantitative values, but relate here to the 2D impurity maps displaying the different analytes, and their qualitative ratios. Investigating elemental ratios on such a small scale has only recently been developed (see Bohleber et al. 2023) and is thus still in its early steps. Further work is needed to apply the same routine to all samples.

**R1:** Line231: from what I see in Figure 1, at 1600 m the mean grain size of ice crystals shows the minimum value in the considered depth interval, with a smoothed mean value of mean grain area (red curve) which is near to 1 mm2. This is not in accordance with what you say in the text, you claim in fact that at this depth ice crystals are larger, but I can't really see this in the graph. The depth of 1600 m corresponds to the LGM, the coldest phase of the last glacial period and also the one where the ice present the highest impurity content. It is well-known that dusty ice presents smaller crystals because the dynamic recrystallization is inhibited by the high number of defects found in the ice lattice. About this you can see for example (Durand et al., 2006). Your data well confirm previous findings, also in relation to the overall increase of grain size with depth, a classic feature observed in deep ice metamorphism. In the light of this, I would change your text, including a brief discussion of how climatic features influence ice microstructure in the EGRIP ice core.
**A:** The presented data in Fig. 1 shows that the mean grain size at 1600 m measured in the discrete samples (black dots) is larger than above and below, which is exactly what we state in line 231. The locally weighted regression line (violet) thus shows a (hardly visible) increase from 1600 m downwards. In general, the grain size in the LGM is very small, we specifically mention the outlier at 1600 m.

We changed the results to: "Mean grain size decreases from 3.6 $mm^2$ at 1340 m depth to 1-2 $mm^2$ by 1800 m (except 2-2.7 $mm^2$ at 1600 m depth) (Fig. 1). The mean grain size is at its minimum between 1500 and 1700 m correlating with the Last Glacial Maximum. Below, values increase and spread with depth and are between 0.8 and 5.2 $mm^2$. Analysed thin sections contain between 471 and 4550 grains.".
We further edited the discussion part and implemented a brief climatic discussion and add the anti-correlation with grayscale data.

**R1:** Line 235-236: I don-t understand why pointing out that your data are unprecedented if you don't explore or discuss them, I would remove this passage.
**A:** Removed.

**R1:** Line 238: what you mean when you say "develop"? Do you mean increase in size? I would rephrase with something like "The increase of ice grain size with depth is faster at EGRIP than NEEM...". I am also asking what you mean ejrm you say that ice metamorphism is faster and more intense at EGRIP because of the location and of the strong dynamic of the glacier at the site. I mean, I understand the point but I think that now a reader who is not aware about ice metamorphism would not understand. I suggest you to rephrase more clearly and add some references about the effect of strain on ice grain size and also of temperature maybe (you are saying that recrystallization is faster at EGRIP than at NEEM also because of the lower elevation and the higher temperature right?).
**A:** We adapted the sentence as suggested. We extend the grain size discussion (see above). A detailed discussion of the processes impacting grain size would need a more detailed investigation, i.e. the analysis of grain size and microstructure on the small scale (i.e. mm-cm). This will potentially be done together with a detailed analysis of the ice fabric, which will enable a thorough discussion of processes affecting grain size and deformation. We however added some more details and reference studies on NEGIS/EGRIP: "The grain size evolution at EGRIP is similar to the NEEM ice core (Montagnat et al., 2014) (Fig. 7), but slightly shifted upwards. This could be related to the different boundary conditions (temperature, elevation) at the drill sites, the impact of extensional deformation and high strain inside NEGIS and the hard shearing in flow direction (Westhoff et al., 2021; Stoll et al., 2021b; Gerber et al., 2022), and the strong dynamic recrystallisation observed at EGRIP (Stoll et al., 2021b). However, the grain size evolution within both cores is still comparable, and the fast-flowing ice within NEGIS does not (yet) intensely affect the grain size in the upper 2121 m (~80% of ice sheet thickness)."

**R1:** Figure7: would it be possible to add at panel a (I think you should always distinguish your figures into panels if they are composed by more than a part) the smoothed curves of the data, this would help appreciating the similarities and differences between the two cores. It would also be nice to include EGRIP isotopic data to be consistent with the comparison between the grain size, are these data available?
**A:** Unfortunately, isotope data from the EGRIP glacial is not available and will only be partly measured. It will not be available before the entire core is drilled and processed (not before winter 2023) and will then probably be published in a dedicated isotope paper.
We adjusted the figure as suggested. We initially refrained from labeling since the isotope data are not ours and give context to the grain size data, thus have to be viewed together. However, panels are now included.

**R1:** Line243-245: this passage is not really well written, I suggest to rephrase or also to delete it.
**A:** Removed.

**R1:** Line 249-250: are you talking about impurities in glacial ice? It is not clear, it seems that you missed to add the term of comparison.
**A:** we added a figure showing the diversity and changed the text to: "In general, mineral variability in glacial ice is slightly lower than what is found in Holocene ice (Fig. A5) (Stoll et al., 2022). This is mostly due to a richer diversity of sulfate minerals found in interglacial ice, especially in the shallowest 900 m."

**R1:** Line250: comparable to what? I would rephrase for example as "In general, mineral variability in glacial ice is slightly lower than what is found in Holocene ice (Stoll et al., 2022). This is mostly due to a richer diversity of sulfate minerals found in interglacial ice."
**A:** Adjusted, see above.

**R1:** Line 252-253: I don't understand this passage, in S^ you identified 69 minerals (I see from Figure 3), here you say that in this sample you could not identify 8 minerals, corresponding to about 10% of total observed minerals. Why are you saying that in this sample "the total amount of different minerals at this depth is higher than identified", I can't get the point.

**A:** This is indeed confusing. The spectra of 8 inclusions could not be identified, the number of different minerals/spectra in sample S6 (not to be confused with the number of identified inclusions) is thus higher than the number of identifies minerals/spectra. As an example, there could be 30 identified inclusions showing only spectra of quartz and feldspar, the number of different minerals would thus be 2. Several unidentified spectra thus indicate a higher diversity of minerals. We refrained from discussing the unidentified spectra and showing the total number of identified mineral groups (see Stoll et al., 2022), because the manuscript is already rather long. We added a new figure to the appendix (Fig. A5) showing this.

To clarify this we changed the sentence: "In S6, 8 spectra could not be identified indicating the presence of more mineral groups than we could classify."

**R1:** Figure 8: I don't think this is the best graph o show the evolution of mineral assemblages over depth (and climatic stages). Spider plots are the best to highlight trends referred to single species (maybe to highlight this it would be nice to add a linear trend to the data), but to have an idea of how the overall mineral assemblage changes with depth, you could prepare pie charts showing mean data about each climatic period (so one chart for the Holocene, one for YD, one for BO, one for the glacial0. In this way it could be possible to really see if the climate and mineral diversity are somehow related, now this is extremely difficult to see. If I interpreted correctly the grey lines are all the minerals, while the highlighted ones refer to the specific minerals referring to single panels. Is this right? You should better explain this in the caption, however I suggest to remove grey lines, it would be clearer to have a single line for each panel, I don't see the reason to show all the minerals in each panel. I have seen that when you did not find a mineral you just skipped the point associated to the considered sample. I suggest to add 0 values in place, in this way it would be possible to better appreciate when the mineral is not present (that is an information). For example if I look at hematite it seems that in the Holocene its concentration is rather constant, but this is only because there are many samples where you did not find hematite that are not reported. I suggest to deeply revise this figure and improve its readability.

**A:** Yes, your interpretation of the figure is correct. We tried different approaches and came up with some changes. We added 0 values to increase readability and lowered the opacity of the grey lines, which we keep to enable a quick relative comparison between the different minerals in one sample. Adding a linear trend, however, resulted in an overcrowded plot. Spider/radar plots also turned out to be confusing and hard to grasp (see https://www.data-to-viz.com/caveat/spider.html for a discussion of spider plots). Nevertheless, we added pie charts for the Holocene, YD, BO, and glacial using the same colour scale to summarise the different mineralogies. Furthermore, we revised the figure caption. We are happy to edit the figure further if required by the editor.

**R1:** Line 254: considering the above comment, I can't really see the decreased mineral diversity in glacial ice according to the way they are now presented, I suggest you to prepare a different type of graph to better discuss this point.

**A:** We deleted the reference to Fig. 8 and added a dedicated diversity index figure (Fig. A5) to the appendix, which is now referred to here.

**R1:** Line 255: I do not agree with your interpretation. It is for sure true that during the glacial periods the atmosphere (and thus the ice) is dustier, but this is not directly related to an increased mineral diversity. From what we know, glacial dust is mostly supplied by specific sources that activate under glacial conditions, so there is a lot of dust but its signature is rather uniform. On the contrary, during interglacials the atmosphere is cleaner because of a suppressed atmospheric dust cycle, so every small dust source can contribute to dust emission without being overwhelmed by powerful glacial dust sources. This has been noted in Antarctica (see for example Baccolo et al., 2018; Delmonte et al., 2020; Gabrielli et al., 2010) but also in Greenland (see Bory et al., 2003; Svensson et al., 2000; Újvári et al., 2022), where dust diversity in the Holocene is higher than in the LGM. I would change accordingly to this discussion.

**A:** We appreciate the input and change this part accordingly; the relative contribution of the different dust sources explains our observations nicely. We change the text to: "Mineral diversity in the glacial is comparably constant and implies a more differentiated trend between the upper 900 m and the rest of the core. Together with data from Stoll et al. (2022), mineral diversity decreases with depth, peaks in the intermediate Holocene and remains relatively constant throughout the last glacial. In glacials, specific dust sources dominate, resulting in a uniform mineralogy signature. They suppress smaller

dust sources, which thus contribute more to the cleaner atmosphere of interglacials, i.e. the Holocene in our samples, resulting in a higher dust diversity as seen in our data. Especially samples from intermediate depths, i.e. the Last Glacial Maximum, do not show a high mineral diversity (Fig. 8) due to overwhelming dust sources as observed in Antarctica (Gabrielli et al., 2010; Baccolo et al., 2018; Delmonte et al., 2020) and Greenland (Bory et al., 2003; Svensson et al., 2000; Újvári et al., 2022). "

**R1:** Line270-274: finding carbonates in glacial ice and not in interglacial one is something that have already been reported and discussed both for Antarctica and for Greenland. In glacial periods the atmosphere is much more dusty, mineral particles are thus less affected by acidic weathering, both during transport and once trapped into the ice. This is because the acidic species present in the aerosols are efficiently neutralized reacting with the huge amount of dust present in the atmosphere. On the contrary during the integlacials the acidic species are more abundant in relative terms, because dust concentration is much less abundant. Therefore, dust that can potentially react with acidic species (such as carbonates) are easily weathered. The final result is that you don't find much carbonates in ice from interglacial periods, while you can find some in glacial ice, as you are showing with your findings. I suggest you to cite the following papers about this and revise the discussion: Eichler et al., 2019; Iizuka et al., 2008; Ohno et al., 2006.
**A:** We agree and change the text accordingly: "Dolomite and calcite occur for the first time in sample 10 and sample 14, respectively, and only at and below intermediate depths, similar to findings by e.g. Ohno et al. (2006); Iizuka et al. (2008); Eichler et al. (2019). The dusty atmosphere in glacial times reduces the acidic weathering of mineral particles (during transport and in the ice) because acidic species in the aerosols are neutralised by reacting with the dust. Hence, dolomite occurs abundantly in sample 10 from the dust-rich Younger Dryas and in lower numbers in the deepest samples".

**R1:** Line284-289: you could compare this result with Baccolo, Delmonte, Di Stefano, et al., 2021, where the only other available hematite record from a polar ice core is presented. Hematite is generally not stable at pH lower than 4 (Schwertmann & Murad, 1983; Zolotov & Mironenko, 2007), so the fact that you find it across the entire core could be used as a proxy for this information. You could also mention this. Since hematite is produced in relatively warm and arid environments (Schwertmann, 1988), its presence is also in accordance with the main source for Greenlandic dust (i.e. arid areas in central Asia). This is another thing to tell here. Again i am asking myself why you did not find any trace of goethite, that should be much more abundant in dust deposited in polar and cold regions (there is some literature about that).
**A:** We added this useful information. It is further interesting that Eichler et al. (2019) found hematite in 129 kyr old EDML ice, the geochemical processes in the ice thus seem to be more complex and might vary from ice core location to ice core location, and potentially in the here investigated cloudy bands. In general, our results agree with Baccolo et al., 2021 since hematite was found in both studies (at least) down to 49 ka old ice. The remaining 540 m of the EGRIP ice core (which partly still has to be drilled) would have to be analysed to compare it with the TALDICE ice showing no more signs of hematite. Further research is thus needed.
We changed the text to:"Hematite (third row in Fig. 8) origins from weathering in soil, banded iron formations or places with standing water and can occur in low abundances together with minerals from igneous or metamorphic rocks. Its origin in warm and arid environments (Schwertmann, 1988) thus follows the principal dust sources for Greenland, the deserts in central Asia. Hematite occurs regularly throughout the core, following its low abundance in rocks but high chemical stability against weathering, with minimum numbers in sample 11 and comparably high numbers in the deepest part of the core (samples 19-24) (Fig. 8). It is not stable under acidic conditions (pH<4) and dissolves (Schwertmann and Murad, 1983; Zolotov and Mironenko, 2007), thus indicating higher pH values throughout the analysed depth regime. Hematite was found in relatively low amounts throughout the Antarctic TALDICE ice core between MIS3 (31-58 ka) and the Holocene (0-11.7 ka) (Baccolo et al., 2021), which agrees with our findings. However, the hematite amount in EGRIP peaks at 37.3 and 39.9 ka (MIS3) instead of MIS2 as in TALDICE (Baccolo et al., 2021). Similar to Eichler et al. (2019), we only found hematite and not precipitated goethite and jarosite (Baccolo et al., 2021)."

**R1:** Line 290: you have never introduced the concept of ice as a geochemical reactor (which actually was introduced not in the paper you are citing (Baccolo, Delmonte, Niles, et al., 2021), but in (Baccolo, Delmonte, Di Stefano, et al., 2021)), introducing this so abruptly is not very clear. You could change to something like "Nitrates and sulfates in the third row in Fig. 8 are minerals, which have been recognized as byproducts of weathering processes that involve dust trapped in deep polar ice".
**A:** We apologize for the wring citation and corrected the reference. We also implemented the

suggestion. Discussing the geochemical reactor concept further would go beyond the scope of this manuscript but should be addressed in follow-up studies focusing solely on chemistry.

**R1:** Line 318-319: it seems that in this passage you are missing a word. It does not sound correct.
**A:** Adjusted to "localisation of Na, Mg, and Sr and widen the analysis to elements"

**R1:** Paragraph 4.3 and 4.4: I found the first part (4.3) a bit wordy. I think this could be greatly shortened. At the end the main point is that combining Raman and LA-ICPMS allows to fully describe how impurities are distributed in deep polar ice, i.e. insoluble crustal elements concentrated in intra-gran inclusions, more mobile and soluble impurities at grain junctions because of ice-crystallization. What is not very clear in these two paragraphs (especially 4.4) is the link between your data (laser ablation one in particular) and cloudy bands. Looking at Figure 5 and 6, where you compare Raman spectrometry, laser ablation and visual stratigraphy, I can't really understand how you link these 3 elements with the presence or absence of cloudy bands. For example you never fully discuss the variability that is observe in results from LA-ICPS. You data clearly show some trends occurring at grain junctions, with some part of the sample that are richer in Fe and other in Ti (see for example S11 in Figure 6), in other cases you have nice accumulation of elements at junctions and suddenly ice much purer or with a higher abundance of intra-grain inclusions (S10, Figure 5). I think that data integration here should be improved.
**A:** We shortened and condensed both sections. We further connect the data better and show that 1) the method combination is very new and 2) how we integrated the data (Section 4.3). The mentioned figures show the mesoscale (visual stratigraphy) as an orientation, while the darker parts of the Raman maps correlate with cloudy bands visible in the visual stratigraphy. Highlighting these parts was an idea, but it made the maps appear very crowded. The LAICPMS data clearly shows the differences in element intensity in and outside cloudy bands. Especially Fig. 5 and S10 in Fig 6 show the substantial chemical differences between cloudy bans and the area around them, thus displaying the linkage of the three methods. However, this study is a starting point to develop an inter-method approach investigating cloudy bands and potentially other features in more detail in the future. In sect. 4.4 we refer several times to explicit examples of chemical data (in the form of figures) supporting our interpretation. These data mainly show thin bright layers with high (solid) impurity concentration and certain minerals localised in layers (Raman + LA-ICP-MS) and darker areas with lower concentration impurity concentration and less mineral localisation. Repeating these results would only lengthen the section discussed in detail in earlier sections; 4.4 synthesises the different data sets, and explaining them again does not seem reasonable here.

**References Reply**

Monika Hanesch, Raman spectroscopy of iron oxides and (oxy)hydroxides at low laser power and possible applications in environmental magnetic studies, *Geophysical Journal International*, Volume 177, Issue 3, June 2009, Pages 941–948, https://doi.org/10.1111/j.1365-246X.2009.04122.x

**References Reviewer**

Baccolo, G., Delmonte, B., Albani, S., Baroni, C., Cibin, G., Frezzotti, M., Hampai, D., Marcelli, A., Revel, M., Salvatore, M. C., Stenni, B., & Maggi, V. (2018). Regionalization of the Atmospheric Dust Cycle on the Periphery of the East Antarctic Ice Sheet Since the Last Glacial Maximum. *Geochemistry, Geophysics, Geosystems*, *19*(9), 3540–3554. https://doi.org/10.1029/2018GC007658

Baccolo, G., Delmonte, B., Di Stefano, E., Cibin, G., Crotti, I., Frezzotti, M., Hampai, D., Iizuka, Y., Marcelli, A., & Maggi, V. (2021). Deep ice as a geochemical reactor: Insights from iron speciation and mineralogy of dust in the Talos Dome ice core (East Antarctica). *The Cryosphere*, *15*(10), 4807–4822. https://doi.org/10.5194/tc-15-4807-2021

Baccolo, G., Delmonte, B., Niles, P. B., Cibin, G., Di Stefano, E., Hampai, D., Keller, L., Maggi, V., Marcelli, A., Michalski, J., Snead, C., & Frezzotti, M. (2021). Jarosite formation in deep Antarctic ice provides a window into acidic, water-limited weathering on Mars. *Nature Communications*, *12*(1), 1. https://doi.org/10.1038/s41467-020-20705-z

Bory, A. J.-M., Biscaye, P. E., & Grousset, F. E. (2003). Two distinct seasonal Asian source regions for mineral dust deposited in Greenland (NorthGRIP). *Geophysical Research Letters*, *30*(4). https://doi.org/10.1029/2002GL016446

Brook, E. J., & Buizert, C. (2018). Antarctic and global climate history viewed from ice cores. *Nature*, *558*(7709), 7709. https://doi.org/10.1038/s41586-018-0172-5

Delmonte, B., Winton, H., Baroni, M., Baccolo, G., Hansson, M., Andersson, P., Baroni, C., Salvatore, M. C., Lanci, L., & Maggi, V. (2020). Holocene dust in East Antarctica: Provenance and variability in time and space. *The Holocene*, *30*(4), 546–558. https://doi.org/10.1177/0959683619875188

Durand, G., Weiss, J., Lipenkov, V., Barnola, J. M., Krinner, G., Parrenin, F., Delmonte, B., Ritz, C., Duval, P., Röthlisberger, R., & Bigler, M. (2006). Effect of impurities on grain growth in cold ice sheets. *Journal of Geophysical Research: Earth Surface*, *111*(F1). https://doi.org/10.1029/2005JF000320

Eichler, J., Weikusat, C., Wegner, A., Twarloh, B., Behrens, M., Fischer, H., Hörhold, M., Jansen, D., Kipfstuhl, S., Ruth, U., Wilhelms, F., & Weikusat, I. (2019). Impurity Analysis and Microstructure Along the Climatic Transition From MIS 6 Into 5e in the EDML Ice Core Using Cryo-Raman Microscopy. *Frontiers in Earth Science*, *7*. https://www.frontiersin.org/articles/10.3389/feart.2019.00020

Gabrielli, P., Wegner, A., Petit, J. R., Delmonte, B., De Deckker, P., Gaspari, V., Fischer, H., Ruth, U., Kriews, M., Boutron, C., Cescon, P., & Barbante, C. (2010). A major glacial- interglacial change in aeolian dust composition inferred from Rare Earth Elements in Antarctic ice. *Quaternary Science Reviews*, *29*(1), 265–273. https://doi.org/10.1016/j.quascirev.2009.09.002

Iizuka, Y., Horikawa, S., Sakurai, T., Johnson, S., Dahl-Jensen, D., Steffensen, J. P., & Hondoh, T. (2008). A relationship between ion balance and the chemical compounds of salt inclusions found in the Greenland Ice Core Project and Dome Fuji ice cores. *Journal of Geophysical Research: Atmospheres*, *113*(D7). https://doi.org/10.1029/2007JD009018

Jouzel, J. (2013). A brief history of ice core science over the last 50 yr. *Climate of the Past*, *9*(6), 2525–2547. https://doi.org/10.5194/cp-9-2525-2013

Langway, C. C. (2008). The history of early polar ice cores. *Cold Regions Science and Technology*, *52*(2), 101–117. https://doi.org/10.1016/j.coldregions.2008.01.001

Ohno, H., Igarashi, M., & Hondoh, T. (2005). Salt inclusions in polar ice core: Location and chemical form of water-soluble impurities. *Earth and Planetary Science Letters*, *232*(1), 171–178. https://doi.org/10.1016/j.epsl.2005.01.001

Ohno, H., Igarashi, M., & Hondoh, T. (2006). Characteristics of salt inclusions in polar ice from Dome Fuji, East Antarctica. *Geophysical Research Letters*, *33*(8). https://doi.org/10.1029/2006GL025774

Sakurai, T., Iizuka, Y., Horikawa, S., Johnsen, S., Dahl-jensen, D., Steffensen, J. P., & Hondoh, T. (2009). Direct observation of salts as micro-inclusions in the Greenland GRIP ice core. *Journal of Glaciology*, *55*(193), 777–783. https://doi.org/10.3189/002214309790152483

Schwertmann, U. (1988). Occurrence and Formation of Iron Oxides in Various Pedoenvironments. In J. W. Stucki, B. A. Goodman, & U. Schwertmann (Eds.), *Iron in Soils and Clay Minerals* (pp. 267–308). Springer Netherlands. https://doi.org/10.1007/978-94-009-4007-9_11

Schwertmann, U., & Murad, E. (1983). Effect of pH on the Formation of Goethite and Hematite from Ferrihydrite. *Clays and Clay Minerals*, *31*(4), 277–284. https://doi.org/10.1346/CCMN.1983.0310405

Svensson, A., Biscaye, P. E., & Grousset, F. E. (2000). Characterization of late glacial continental dust in the Greenland Ice Core Project ice core. *Journal of Geophysical Research: Atmospheres*, *105*(D4), 4637–4656. https://doi.org/10.1029/1999JD901093

Újvári, G., Klötzli, U., Stevens, T., Svensson, A., Ludwig, P., Vennemann, T., Gier, S., Horschinegg, M., Palcsu, L., Hippler, D., Kovács, J., Di Biagio, C., & Formenti, P. (2022). Greenland Ice Core

Record of Last Glacial Dust Sources and Atmospheric Circulation. *Journal of Geophysical Research: Atmospheres*, *127*(15), e2022JD036597. https://doi.org/10.1029/2022JD036597

Zolotov, M. Y., & Mironenko, M. V. (2007). Timing of acid weathering on Mars: A kinetic-thermodynamic assessment. *Journal of Geophysical Research: Planets*, *112*(E7). https://doi.org/10.1029/2006JE002882

---

## Author Comment (AC2)

**Comment on tc-2022-250**

Anonymous Referee #2

Referee comment on "Chemical and visual characterisation of EGRIP glacial ice and cloudy bands within" by Nicolas Stoll et al., The Cryosphere Discuss., https://doi.org/10.5194/tc-2022-250-RC2, 2023

**R2:** Stoll et al. provide an interesting insight into the physical and chemical characterization of cloudy bands in glacial ice. The authors use a combination of visual stratigraphy line scanner, fabric analyser, microstructure mapping, Raman spectroscopy, and laser ablation inductively coupled plasma mass spectrometry 2D impurity imaging to classify the cloudy bands from the EGRIP ice core and also studied the localization and mineralogy of several micro-inclusions at different depths. Cloudy bands have been studied from the early days of ice cores research, but insights into their formation mechanism were severely lacking, which is addressed in this work. This study provides a great starting point for a better understanding of the different structural features of glacial ice.

The manuscript presents a sizable amount of valuable data. The methodology used in the study is robust and provides a high level of detail on the chemical and visual characteristics of the ice and cloudy bands. The figures and tables are well-presented and provide clear visual representations of the results. The authors also provide a detailed analysis of the results. However, I feel some results are presented but not well-discussed in the manuscript (see specific comments). While I agree it's impossible to discuss everything in one manuscript, it also doesn't make sense to present data and not discuss it sufficiently. The overall idea of the work and the data present are unique and relevant to The Cryosphere and the glaciological community. Therefore, the manuscript will be ready for publishing after a revision addressing the major issues.

**A:** We thank the reviewer for constructive feedback, which certainly improves the quality of the manuscript. We implement the suggestions and worked on the discussion, especially on section 4.3 describing the data integration and interpretation.

**Specific comments:**

**R2:** Line 2: "Glacial period" would be more apt term here.
**A:** Changed to "glacial periods".

**R2:** Line 6: Replace "almost" with "approximately". Approximately works better when you mention numbers.
**A:** Changed as suggested.

**R2:** Line 8: Replace "found minerals" with "minerals found".
**A:** Changed as suggested.

**R2:** Line 9: If I understand correctly, these minerals are observed rarely, but one might confuse the phrase "rare minerals" with "rare earth minerals". This can be better written as "Rutile, anatase, epidote, titanite, and grossular are rarely observed/found".
**A:** Changed as suggested.

**R2:** Line 10: Replace "with" with "at". Add "present" or "occur" with "mainly"; otherwise, the sentence sounds a little incomplete.

**A:** Changed as suggested.

**R2:** Line 10-11: Maybe rephrase this as "Whereas, dust-related analytes, such as Al, Fe, and Ti are located in the grain interior, forming clusters of insoluble impurities". Then you could continue with your next statement and say that the cloudy bands are distinguishable.
**A:** Changed as suggested.

**R2:** Line 23: What do you mean by "grain size"? Insoluble particle grain size or ice crystal size? Please clarify this.
**A:** Changed to "Considering different polar ice cores, most of the physical and chemical properties of ice and its impurities vary, depending on several parameters that are different at each drilling site.". We further clarify grain size later in the text.

**R2:** Line 34: What does "typical" mean? Do you mean a characteristic description of a cloudy band? If so, then write clearly. Why not just write "cloudy band have been discussed for a variety of reasons, ranging from climatic to deformation aspects".
**A:** Changed to: "Cloudy bands vary in thickness, brightness, and shape and are thus hard to constrain (Winstrup et al., 2012), but they were discussed for a variety of reasons, ranging from climatic to deformation aspects.

**R2:** Line 36-38: This part is a bit hard to read. Maybe this can better written as " Svensson et al (2005) compared the brightness intensity values derived from visual stratigraphy with different records from continuous flow analysis (CFA) and showed that the brightness variations in cloudy bands mostly match the seasonal cycles of others tracers.
**A:** Changed to: ". Svensson et al. (2005) show that, in most cases, the brightness variations of visual stratigraphy and cloudy bands match the seasonal cycles of tracers, especially of dust, derived by continuous flow analysis (CFA) (Fig. 5 in Svensson et al., 2005)".

**R2:** Line 61: Replace "and the chemistry" with ", chemistry,".
**A:** Changed as suggested.

**R2:** Line 64-65: Do you means "few tens of microns"?
**A:** Yes, changed as suggested.

**R2:** Line 78: Detele "again"
**A:** Changed as suggested.

**R2:** Line 89: Since you study both micro-inclusions and ice fabric, its best to clarify what you mean by "grain size". Maybe use separate terms for both, like particle size (micro- inclusions) and crystal size(ice crystals) and avoid using the terms interchangeably.
**A:** We clarify this by adding "(in this study grain size refers to the ice crystal)".

**R2:** Line 90: Is there a specific reason why the samples are 55cm long?
**A:** The length of 55 cm has historical/logistical reasons, i.e. the Danish ice cores boxes are 55 cm long. DEP, ECM, and the line scanner are the only techniques analysing the (uncut) 165 cm sections. However, they still separate the data in three 55 cm sections for easier comparison with other data sets.

**R2:** Line 101: Remove all "e.g."
**A:** Changed as suggested.

**R2:** Line 104: In the dark-field setup, the camera records the light that is "scattered" by the micro-inclusions and not "reflected" (Svensson et al., 2005). The only reflected light recorded is from the ice core periphery. Please correct it throughout the manuscript.
**A:** Changed to "Slabs of ice were polished from both sides and illuminated from below ("dark field" imaging). The light is scattered by solid impurities, fractures, and bubbles and directed back into the camera making them visible. In glacials, the main scattering objects are cloudy bands and fractures."

**R2:** Line 118-119: Why not show examples of what you group as "unknown"? From what I understand, if the images are too dark or layers are too thin, you cannot be sure that the features are cloudy bands. Then why consider them. Moreover, in line 110 you say that you don't investigate deformed bands, so why do you consider them "unknown"? If the bands are classified, that would mean that they are investigated! If that's not the case, then clarify the same in the text.
**A:** We explain this in detail in an answer to the first review report and an include examples in the Appendix (Fig. A2).

**R2:** Line 120: Just because the bands are present in three ice cores, is it prudent to say that the band types are representative of Greenland? I'm not at all against a claim like this, so it's up to the authors to decide.
**A:** We discussed this wording before submission and recognized the difficulty in generalizing. Unfortunately, there is not much continuous visual stratigraphy data available and such a claim is thus tricky to discuss, but it is important to state that our results are transferable to other ice cores. We add "probably" to make the statement less absolute.

**R2:** Line 123: Cloudy bands can be a few milimeters thick too (you too mention this in line 31). So, it's better to write this as "mm to cm-scale" instead of "cm-scale". Change "analyze" to "analyse". You mostly use the British variant, so stick to one through the manuscript.
**A:** Changed as suggested.

**R2:** Line 146: Replace "before" with "earlier".
**A:** Changed as suggested.

**R2:** Line 152-153: What is the grain size variability within a sample (mean +- std)? Would it be possible to show this in figure 1 with a patch in the background representing the range? That would give a better insight into the range of variability in grain sizes with depth.
**A:** How to handle, i.e. measure, display and evaluate grain size data in the best way remains difficult, and so far, no standard procedure has been established in glaciology. Our method enables precise measurement of the entire (2D) grain area by evaluating single pixels, which is much more robust than classical diameter measurements. The variability can be large, especially when cloudy bands with fine grains and much larger grains occur in the same sample. Due to the high numbers of grains and thus good statistics, displaying the sample means remains our preferred option while preserving readability. Ongoing work investigates the EGRIP microstructure in more detail, including the shape preferred orientation (SPO) of ice grains and how grain size varies on the small scale, i.e. within samples. We thus prefer to avoid additional data (and discussion) in this manuscript (since it is already packed), which will be the scope of a specific manuscript.

**R2:** Line 157-158: I believe this is your observation from the EGRIP grayscale data. Why do you then cite Rasmussen et al., 2006?
**A:** This is indeed confusing, we here refer to the timing of the stadials and interstadials as described in Rasmussen et al., 2006. We delete it to avoid confusion.

**R2:** Line 158-160: I don't understand the points of writing all this here? What's the relevance? Why don't you better describe your grayscale data here, explaining the overall grayscale variability and its co-variability with other causal factors. If you feel there's isn't much to describe here or you don't want to go in-depth on the factors affecting grayscale variability, remove this part.

**A:** We agree and shortened this part and combined it with the subsection below. Grayscale is used to classify cloudy bands, which we now mention. Furthermore, grayscale/brightness and grain area are often anti-correlated indicating the impact of solid particles on grain growth. This is displayed in Fig. 1 and now mentioned in the grain size discussion.

**R2:** Line 166: Is there a specific reason to choose 55 cm sections? The line scans were done for ~1.65 m section (figure A1) then why 55 cm sections are used?

**A:** See answer above. To enable intercomarability between different techniques it is standard to work with 55 cm sections.

**R2:** Line 172: No need to capitalize stadials and interstadials. Change them to sentence case throughout.

**A:** Changed as suggested.

**R2:** Line 180: "Carbonaceous particles" is a very vague term when you are identifying minerals. Its rather confusing to me. Could you explain this a bit more clearly what these carbonaceous particle are?

**A:** Changed to: "t further distinguished particles bearing carbon (from here on carbonaceous particles)".

**R2:** Line 193: I suggest replacing "number of different" with "diversity of".

**A:** Changed to "Shallow samples (S1-S6) have a diversity of 9 to 14 minerals, while deeper samples (S7-S13) show a variety of 6 to 10 minerals per sample.".

**R2:** Line 195: Use the term "rare mineral" carefully. It can mislead the readers.

**A:** Changed to "rarely observed mineral".

**R2:** Line 216: Delete "especially", "the ".

**A:** Changed as suggested.

**R2:** Line 217: "rarely" would be a better word than "infrequently"

**A:** Changed as suggested.

**R2:** Line 238: Can you expand this line, so that it is understandable to everyone as to what all factors are different between EGRIP and NEEM site, and how they change the grain size evolution?

**A:** Changed to: "The grain size evolution at EGRIP is similar to the NEEM ice core (Montagnat et al., 2014) (Fig. 7), but slightly shifted upwards. This could be related to the different boundary conditions (temperature, elevation) at the drill sites, the impact of extensional deformation and high strain inside NEGIS and the hard shearing in flow direction (Westhoff et al., 2021; Stoll et al., 2021b; Gerber et al., 2022), and the strong dynamic recrystallisation observed at EGRIP (Stoll et al., 2021b). However, the grain size evolution within both cores is still comparable, and the fast-flowing ice within NEGIS does not (yet) intensely affect the grain size in the upper 2121 m (~80% of ice sheet thickness)."

**R2:** Line 243-245: Is it really required? This is more like a concluding remark!

**A:** Deleted.

**R2:** Line 319: Add "and" before "widen". Sounds incomplete without it.
**A:** We changed the entire paragraph, see new manuscript.

**R2:** Line 348: Why do you suddenly shift to third person? Replace "The authors" with "We".
**A:** This refers to the study by Svensson et al. (2005) mentioned in the previous sentence. We change "the authors" to "Svensson et al. (2005)".

**R2:** Line 355-357: It's a very confusing sentence. Maybe replace with "Wet deposition would lead to increasing layer thickness and thus resulting in a thick cloudy band".
**A:** Changed as suggested.

**General questions:**

**R2:** You use terms like "bright layer", "dark layer", "medium bright layer", and "moderately bright layer" in the manuscript. How do you differentiate between them? Is there a quantifiable value for these layers? If so, can you please explain how you came to define these layers as such?
**A:** The terms refer to grayscale pixel values ranging from 0 to 255, creating the following boundary conditions for the grouping: Below 100-dark, 100 to 150-medium dark, around 150-medium, 150 to 200-medium bright, 200 to 255-bright. We add this information in section 3.2, which now combines grayscale and cloudy band types.
The boundaries are not fixed, as the brightnesses are relative. A medium bright layer can be significant inside a thick cloudy band, although it does not satisfy the definition of exceeding a pixel value of 200.

**R2:** Do you need to use gridlines in the plots? Some figures are very hard to understand because of the closely packed gridlines.
**A:** Gridlines help identify absolute values. We checked the five figures with gridlines and do not think they make them harder to read. With them, it is easier to get quantitative data, which is the central message of these plots. We think gridlines are set at reasonable distances (e.g. per sample/every 5 ka/every 12.5%). We leave this decision to the editor and are happy to reduce the gridlines of specified plots if requested.

**Reference**

Svensson, A., Nielsen, S. W., Kipfstuhl, S., Johnsen, S. J., Steffensen, J. P., Bigler, M., Ruth, U., and Röthlisberger, R.: Visual stratigraphy of the North Greenland Ice Core Project (NorthGRIP) ice core during the last glacial period, Journal of Geophysical Research, 110, NA-NA, 2005.